# The Mac1 ADP-ribosylhydrolase is a therapeutic target for SARS-CoV-2

Rahul K Suryawanshi[1†‡], Priyadarshini Jaishankar[2†], Galen J Correy[3†], Moira M Rachman[2†], Patrick C O'Leary[4†], Taha Y Taha[1,5†], Yusuke Matsui[1†], Francisco J Zapatero-Belinchón[1], Maria McCavitt-Malvido[1], Yagmur U Doruk[4], Maisie GV Stevens[4], Morgan E Diolaiti[4], Manasi P Jogalekar[4], Huadong Chen[4], Alicia L Richards[5,6,7], Pornparn Kongpracha[5,6,7], Sofia Bali[3], Mauricio Montano[1], Julia Rosecrans[1], Michael Matthay[7], Takaya Togo[2], Ryan L Gonciarz[2], Saumya Gopalkrishnan[5], R Jeffrey Neitz[2,8], Nevan J Krogan[3,5,7], Danielle L Swaney[5,6,7], Brian K Shoichet[2], Melanie Ott[1,9,10*], Adam R Renslo[2*], Alan Ashworth[4*], James S Fraser[3*]

[1]Gladstone Institute of Virology, Gladstone Institutes, San Francisco, United States; [2]Department of Pharmaceutical Chemistry, University of California San Francisco, San Francisco, United States; [3]Department of Bioengineering and Therapeutic Sciences, University of California San Francisco, San Francisco, United States; [4]Helen Diller Family Comprehensive Cancer Center, University of California San Francisco, San Francisco, United States; [5]Quantitative Biosciences Institute (QBI), University of California San Francisco, San Francisco, United States; [6]Department of Cellular and Molecular Pharmacology, University of California San Francisco, San Francisco, United States; [7]Data Science and Biotechnology Institute, Gladstone Institutes, San Francisco, United States; [8]Small Molecule Discovery Center, University of California San Francisco, San Francisco, United States; [9]Department of Medicine, University of California San Francisco, San Francisco, United States; [10]Chan Zuckerberg Biohub-San Francisco, San Francisco, San Francisco, United States

*For correspondence:
melanie.ott@gladstone.ucsf.edu (MO);
adam.renslo@ucsf.edu (ARR);
alan.ashworth@ucsf.edu (AA);
jfraser@fraserlab.com (JSF)

†These authors contributed equally to this work

Present address: ‡Laboratory of Neurological Infections and Immunity, Division of Intramural Research, National Institute of Allergy and Infectious Diseases, National Institutes of Health, Rocky Mountain Laboratories, Hamilton, Montana, United States, Hamilton, United States

## eLife Assessment

This **important** study presents the development of a novel inhibitor for SARS-CoV-2 Mac1 that has potential utility both as an antiviral therapeutic and as a tool for probing the molecular mechanisms by which infection-induced ADP-ribosylation triggers robust host antiviral responses. Though minor gaps in understanding the compound's precise molecular mechanism of action and its ability to target Mac1 from other coronaviruses remain, the evidence for its effects on SARS-CoV-2 in relevant biological models is **compelling**.

**Abstract** Severe acute respiratory syndrome coronavirus 2 (SARS-CoV-2) continues to pose a threat to public health. Current therapeutics remain limited to direct-acting antivirals that lack distinct mechanisms of action and are already showing signs of viral resistance. The virus encodes an ADP-ribosylhydrolase macrodomain (Mac1) that plays an important role in the coronaviral life cycle by suppressing host innate immune responses. Genetic inactivation of Mac1 abrogates viral replication in vivo by potentiating host innate immune responses. However, it is unknown whether this can be achieved by pharmacologic inhibition and can therefore be exploited therapeutically. Here, we report a potent and selective lead small molecule, AVI-4206, that is effective in an in vivo model of SARS-CoV-2 infection. Standard cellular models indicate that AVI-4206 has high target engagement and can weakly inhibit viral replication in a gamma interferon- and Mac1 catalytic activity-dependent

manner. However, a stronger antiviral effect for AVI-4206 is observed in human airway organoids and peripheral blood monocyte-derived macrophages. In an animal model of severe SARS-CoV-2 infection, AVI-4206 reduces viral replication, potentiates innate immune responses, and leads to a survival benefit. Our results provide pharmacological proof of concept that Mac1 is a valid therapeutic target via a novel immune-restoring mechanism that could potentially synergize with existing therapies targeting distinct, essential aspects of the coronaviral life cycle. This approach could be more widely used to target other viral macrodomains to develop antiviral therapeutics beyond COVID-19.

## Introduction

Severe acute respiratory syndrome coronavirus 2 (SARS-CoV-2) continues to be a major threat to public health. Despite the approval of several biologic and small molecule therapeutics, there is an urgent need for new small molecule antivirals with distinct mechanisms of action to overcome potential resistance to existing agents (*von Delft et al., 2023*). While most antivirals target an essential aspect of viral entry or replication, a potential avenue for new antivirals with alternative mechanisms is to target viral proteins that act to blunt the host immune response (*Minkoff and tenOever, 2023*). For example, SARS-CoV-2 has evolved multiple mechanisms to evade and counter interferon signaling (*Kim and Shin, 2021*). The viral proteins involved in such evasion would be valuable drug targets if their inhibition renders the host immune response sufficient to control virus replication and reduce disease severity.

The macrodomain (Mac1) of non-structural protein 3 (NSP3) in SARS-CoV-2 is one such target that plays an antagonistic role to the host interferon response (*Schuller et al., 2023*). Macrodomains are found across the tree of life and catalyze the hydrolysis of ADP-ribose covalent modifications on protein side chains (*Dasovich and Leung, 2023*). Viral macrodomains are found in alphaviruses, hepatitis E virus, and many betacoronaviruses (*Leung et al., 2022*) and in some systems, like murine hepatitis virus, their activity can be essential for viral replication (*Voth et al., 2021*). While SARS-CoV-2 bearing either catalytically inactivating point mutations (*Taha et al., 2023a*) or deletion of the Mac1 domain (*Alhammad et al., 2023*) have minor phenotypes in cell culture, their replication is profoundly attenuated in animal models. This discordance likely reflects the inability of cellular models to recapitulate the complex intercellular and systemic signaling required for proper viral–host immune interactions. The underlying mechanism of action results from the enzymatic activity of Mac1, which counters the wave of ADP-ribosylation that is catalyzed by poly-adenosine diphosphate-ribose polymerase (PARP) proteins during the interferon response (*Kar et al., 2024*; *Parthasarathy et al., 2024*; *Kerr et al., 2023*). While the critical proteins and sites modified by interferon-induced PARPs are not fully characterized, the inhibition of Mac1 should allow ADP-ribosylation and the resulting downstream signaling to persist (*Kar et al., 2024*). Indeed, multiple interferon genes are down-regulated upon infection with wild-type SARS-CoV-2 relative to a Mac1-deficient mutant, consistent with the hypothesis that antiviral interferon signaling could be productively enhanced by Mac1 inhibition (*Taha et al., 2023b*; *Alhammad et al., 2023*).

We (*Gahbauer et al., 2023*; *Schuller et al., 2021*), and others (*O'Connor et al., 2023*; *Schuller et al., 2023*; *Wazir et al., 2024*), have previously developed inhibitors of Mac1 with activity in vitro. However, the therapeutic hypothesis that pharmacological Mac1 inhibition would restore host immune responses and lead to a survival benefit after SARS-CoV-2 infection has not yet been tested. Here, we build on our experimental fragment (*Schuller et al., 2021*) and virtual screening approach (*Gahbauer et al., 2023*) with medicinal chemistry, to develop a potent lead compound, AVI-4206, that engages Mac1 in cellular models and has suitable pharmacological properties to test antiviral efficacy in vivo. In an animal model of SARS-CoV-2 infection, AVI-4206 reduces viral replication, restores an interferon response, and leads to a survival benefit. Therefore, our results validate Mac1 as a therapeutic target via a novel immune-restoring mechanism that could synergize with existing therapies targeting essential aspects of viral replication. The approach could be more widely used to target other macrodomains in viruses beyond SARS-CoV-2.

# Results

## Optimization of in vitro potency against the SARS-CoV-2 macrodomain

Previously, we described two novel Mac1 inhibitors, AVI-92 and AVI-219 (*Gahbauer et al., 2023*), which evolved from fragment screening and virtual screening hits, respectively. Their potency was determined using an ADPr-conjugated peptide displacement-based homogeneous time-resolved fluorescence (HTRF) assay (*Figure 1*). The superposition of the Mac1 crystal structures in complex with both leads inspired a parallel approach to optimization, which was supported by additional high-resolution X-ray structures of the complexes (*Figure 1A, E*, *Supplementary file 1*, *Figure 1—figure supplement 1*). First, we generated a merged compound that used the urea function of AVI-92 in the more lead-like AVI-219 scaffold, thus avoiding the phenolic and carboxylate functionalities present in AVI-92. Indeed, the X-ray structure of the resulting complex between Mac1 and AVI-4051 shows that it preserves and favorably orients the two hydrogen bonding contacts with the carboxylate of Asp22 and exhibits a ~fivefold lower $IC_{50}$ value as compared to AVI-219 in the HTRF assay (*Figure 1B, E*). Further structure–activity relationship (SAR) studies revealed a strong preference for urea (e.g., AVI-1500, $IC_{50}$ of ~120 nM) over acetamide (AVI-1501) or carbamate (AVI-3367) in productively engaging Asp22 (*Figure 1B, E*).

At the other end of the adenosine site, we explored a subsite that is engaged by the pyrrolidinone ring of AVI-219 with the carbonyl function involved in hydrogen bonding with the backbone amides of Phe156/Asp157 (*Figure 1C, E*). In a parallel SAR effort, we had explored this same subsite using a shape-based virtual screening approach called FrankenROCS and noted a hydrophobic patch opposite and below the more polar backbone amides (*Correy et al., 2025*). Leveraging this information, and seeking to enhance hydrophobic contact with these residues, we explored substitutions of the pyrrolidinone ring at C-5. While substituents as large as phenyl were tolerated (AVI-3762 and AVI-3763), these analogs showed reduced ligand efficiency compared to AVI-219. By contrast, a methyl group at C-5 in either stereochemical configuration (AVI-3764 and AVI-3765) improved potency and ligand efficiency. Introducing two methyl groups at C-5 afforded the achiral, gem-dimethyl pyrrolidinone AVI-4636 with an impressive fivefold improvement in potency ($IC_{50}$ of ~200 nM) and enhanced ligand efficiency compared to AVI-219 (*Figure 1C, E*).

Ultimately, combining the ethyl urea side chain of AVI-1500 with the gem-dimethyl pyrrolidinone of AVI-4636 produced AVI-4206, the most potent Mac1 inhibitor identified from this series, with an $IC_{50}$ value of ~20 nM (*Figure 1A, E*). This potency approaches the floor of our HTRF assay, which uses 12 nM of enzyme, and indicates that AVI-4206 is at least ~25- and 60-fold more potent than the AVI-92 and AVI-219 starting points, respectively. The high-resolution co-crystal structure of AVI-4206 confirmed that the desired interaction elements and conformations were maintained from the separate optimization paths (*Figure 1E*). To confirm that the high-affinity binding of AVI-4206 was reflected in inhibition of Mac1 catalytic activity, we used auto ADP-ribosylated PARP10 and a coupled NudT5/AMP-Glo assay to measure ADP-ribose released by the enzymatic reaction (*Kasson et al., 2021*; *Voorneveld et al., 2018*; *Figure 1—figure supplement 2*). This assay demonstrated AVI-4206 potently inhibits Mac1 with an $IC_{50}$ of 64 nM (*Figure 1D*).

## AVI-4206 engages Mac1 in cells with high specificity

Having discovered AVI-4206 as a potent inhibitor of Mac1, we next determined whether this compound could enter cells and bind to Mac1 in this context. Therefore, to assess cellular target engagement, we developed a nanoluciferase-based CEllular Thermal Shift Assay (CETSA-nLuc) assay (*Martinez et al., 2018*; *Sanchez et al., 2022*) to measure thermal stabilization of Mac1 upon compound binding. A549 cells transiently expressing a HiBiT- and FLAG-tagged Mac1 protein were treated with compounds for 1 hr and then incubated across a gradient of temperatures. After heat exposure, cells were lysed and incubated with LgBiT protein which binds to soluble HiBiT-Mac1 protein reconstituting nanoluciferase and producing a luminescent signal. We observed that the $T_{agg}$ (the temperature at which 50% of protein is soluble) shift for compounds at 10 µM mirrored the affinities measured by the HTRF assay, suggesting a dominant role for Mac1 affinity, rather than bioavailability, or another factor, in determining target engagement in cells (*Figure 2A*). Furthermore, we observed a dose-dependent shift in $T_{agg}$ by AVI-4206, with a marked ~10°C shift in cells treated with 10 µM of compound compared to DMSO-treated control cells (*Figure 2B*). The observations were also validated by Western blotting with a FLAG-specific antibody (*Figure 2—figure supplement 1*). Encouragingly, we were also able

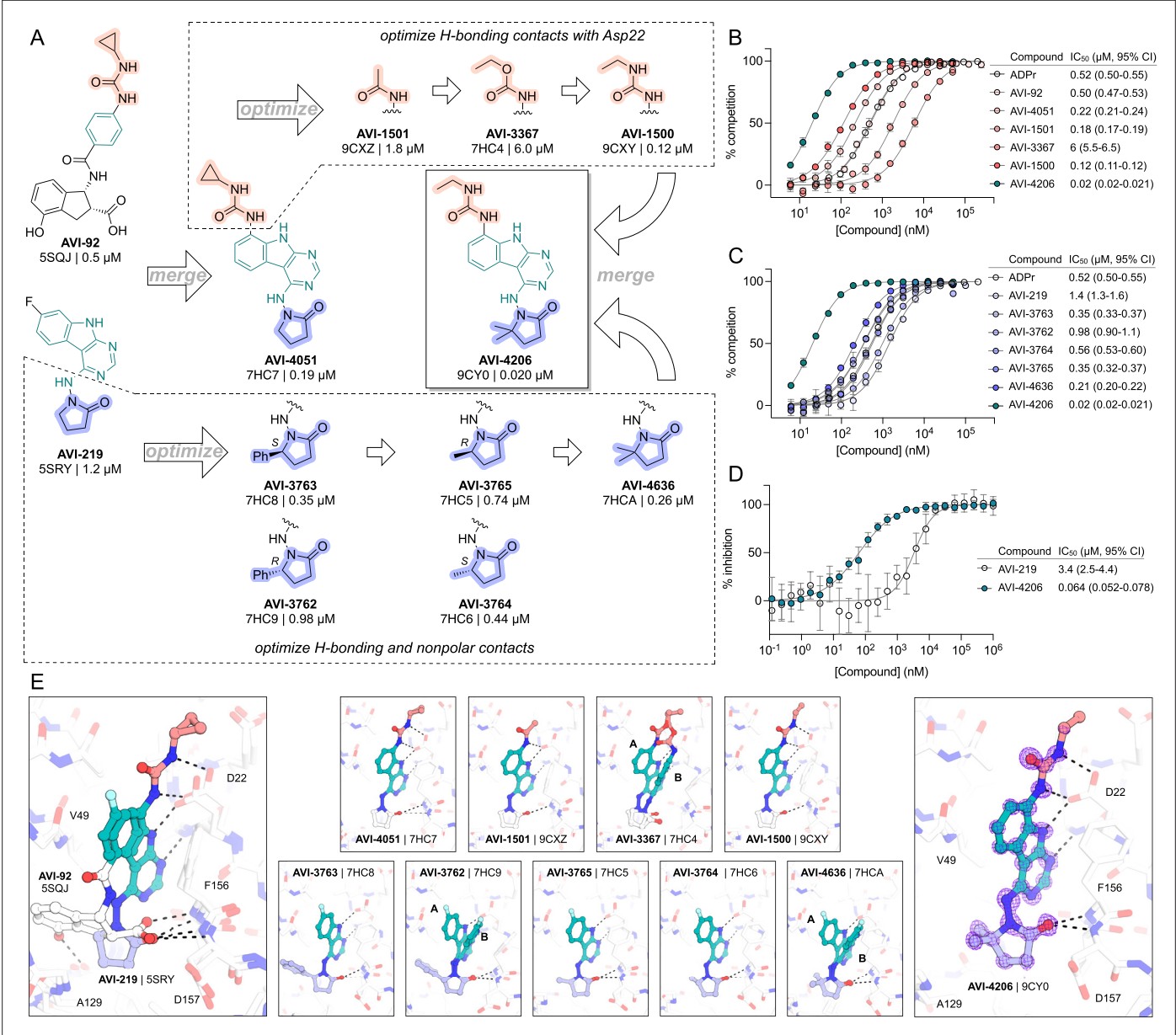

**Figure 1.** Iterative structure-based design and optimization of AVI-4206 activity against Mac1. (**A**) Evolution of the early lead AVI-219 to AVI-4206 by introducing and optimizing urea functionality as found in AVI-92 to contact Asp22 and introducing geminal dimethyl substitution of the pyrrolidinone ring. Homogeneous time-resolved fluorescence (HTRF)-based $IC_{50}$ values from (**B**) and (**C**), and PDB codes from (**E**) are indicated. HTRF-based dose–response curves showing peptide displacement of an ADPr-conjugated peptide from Mac1 by compounds from the urea (**B**) and the pyrrolidinone ring (**C**) optimization paths. Data is plotted as % competition mean ± SD of three technical replicates. Data were fitted with a sigmoidal dose–response equation using non-linear regression and the $IC_{50}$ values are quoted with 95% confidence intervals. (**D**) Mac1 catalytic activity dose–response curve for indicated compounds. Data is plotted as % inhibition mean ± SD of four technical replicates. $IC_{50}$ values are quoted with 95% confidence intervals. (**E**) X-ray structures indicating conserved interactions during the optimization path from AVI-92 and AVI-219 (left) to AVI-4206 (right). Structures of compounds from the urea and the pyrrolidinone ring optimization paths are presented in the top and bottom middle panels, respectively. Multiple ligand conformations were observed for AVI-3367, AVI-3762, and AVI-4636 (labeled A and B). The $F_O - F_C$ difference electron density map calculated prior to ligand modeling is shown for AVI-4206 (purple mesh contoured at 5 σ). Electron density maps used to model ligand other ligands are shown in *Figure 1—figure supplement 1*.

The online version of this article includes the following figure supplement(s) for figure 1:

**Figure supplement 1.** X-ray density for ligand modeling.

**Figure supplement 2.** AVI-4206 and AVI-219 inhibition of Mac1 determined using auto-mono-ADP-ribosylated PARP10 as a substrate.

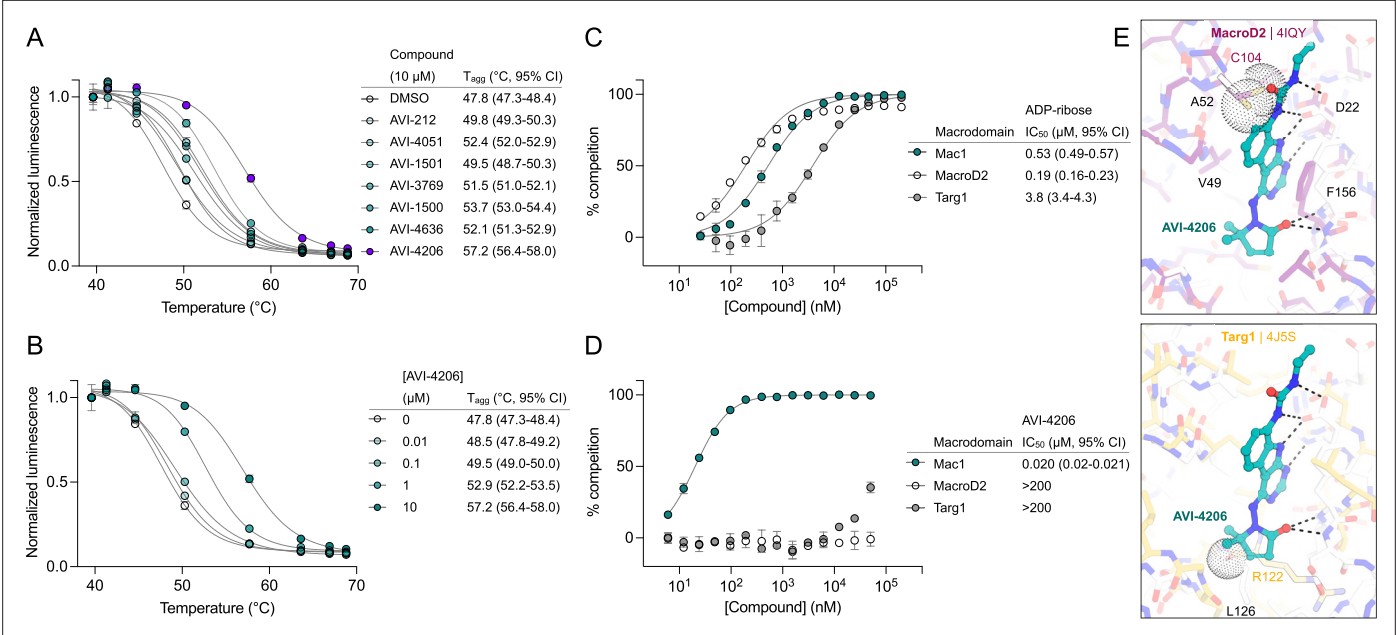

**Figure 2.** AVI-4206 engages Mac1 with high potency and selectivity in cells. (**A**) CETSA-nLuc shows differential Mac1 stabilization after treatment of A549 cells with 10 μM of indicated compounds. Data are presented as mean ± SD of two technical replicates. Data were fitted with a sigmoidal dose–response equation using non-linear regression (gray line) and the $T_{agg}$ values are quoted with 95% confidence intervals. (**B**) CETSA-nLuc shows a dose-dependent thermal stabilization of Mac1 after treatment of A549 cells with increasing concentrations of AVI-4206. Data are presented as mean ± SD of two technical replicates. Homogeneous time-resolved fluorescence (HTRF)-based dose–response curves showing displacement of an ADPr-conjugated peptide from indicated proteins by ADP-ribose (**C**) or AVI-4206 (**D**). ADP-ribose was used as a positive control. Data are presented as mean ± SD of three technical replicates. $IC_{50}$ values are quoted with 95% confidence intervals. (**E**) Structural modeling of MacroD2 (top, PDB code 4IQY) and Targ1 (bottom, PDB code 4J5S) showing design elements that prevent AVI-4206 cross-reactivity. The atoms of clashing residues (Cys140 in MacroD2, Arg122 in Targ1) are shown with a dot representation. The ADP-ribose present in both human macrodomain structures has been omitted for clarity.

The online version of this article includes the following source data and figure supplement(s) for figure 2:

**Figure supplement 1.** AVI-4206 increases thermal stability of Mac1 in cells.

**Figure supplement 1—source data 1.** Labeled full gel of CETSA-WB shows thermal stabilization of FLAG-tagged Mac1 protein after treatment of A549 cells with 10 μM of AVI-4206.

**Figure supplement 1—source data 2.** Unlabeled full gels of CETSA-WB show thermal stabilization of FLAG-tagged Mac1 protein after treatment of A549 cells with 10 μM of AVI-4206.

**Figure supplement 2.** Alignment of AVI-4206-bound SARS-CoV-2 Mac1 with diverse macrodomains suggests the origin of AVI-4206 selectivity.

**Figure supplement 3.** PARP14 macrodomain 1 activity is not inhibited by AVI-4206.

**Figure supplement 4.** Thermal proteome profiling in A549 cellular lysates.

to adapt our CETSA assay for SARS-1 and MERS macrodomain proteins and find that AVI-4206 can shift the melting temperature of both proteins, albeit to a lesser degree than that seen for Mac1 (*Figure 2—figure supplement 1*).

After confirming Mac1 target engagement in cells, we next tested the selectivity of AVI-4206 for Mac1 over two human macrodomains, Targ1 and MacroD2. In an adapted HTRF assay, both human proteins bind to ADP-ribose in the same low-μM range as Mac1 (*Figure 2C*), AVI-4206 does not bind appreciably to either protein in this assay (*Figure 2D*). The selectivity of AVI-4206 for the active site of Mac1 can be rationalized by the presence of larger residues at key positions in the binding pocket in the human orthologs. In Targ1, Cys104 occupies the analogous position to Ala52 of Mac1, leading to a putative clash with the urea moiety (*Figure 2E*). Similarly, in MacroD2, Arg122 occupies the analogous position to Leu 126; both the larger arginine side chain and accompanying backbone shift are predicted to clash with the gem-dimethyl of AVI-4206 (*Figure 2E*). Modeling of other diverse macrodomains, including those within human PARP9 and PARP14, further suggests that AVI-4206 is selective for Mac1 (*Figure 2—figure supplement 2*). Indeed, a catalytically active macrodomain-containing construct of PARP14 (MD1–MD2) is not inhibited by AVI-4206 (*Figure 2—figure supplement 3*). Due

to the shared adenosine motif in the substrates for macrodomains and kinases, and the therapeutic importance of protein kinases, we assessed AVI-4206 at 10 µM against a panel of diverse kinases and found no inhibition >35% (*Supplementary file 2a*). Lastly, we used mass spectrometry-based thermal proteome profiling (TPP) (*Savitski et al., 2014*) to evaluate the selectivity of AVI-4206 against a complex proteome. We added 50 nM recombinant Mac1 protein into cellular lysates from A549 cells that were treated either with DMSO or with 100 µM of AVI-4206. We find that Mac1, but no native protein from the A459 lysate, displays a statistically significant shift in melting temperature (3.02°C, adjusted p value = 0.045) (*Figure 2—figure supplement 4*). While this assay may not be sensitive to detection of proteins with low abundance or low thermal shift upon ligand binding, collectively, these results indicate that AVI-4206 can cross cellular membranes and engage with high specificity for Mac1.

## AVI-4206 displays limited efficacy in simple cellular models

To determine whether AVI-4206 can inhibit viral replication in cellular models of SARS-CoV-2 infection, we treated IFN-deficient Vero cells stably expressing TMPRSS2 (Vero-TMPRSS2) and IFN-competent A549 cells stably expressing high levels of ACE2 (A549-ACE2[h]) with AVI-4206 and infected them with an mNeon reporter SARS-CoV-2 WA1 strain (*Xie et al., 2020*). We observed that treatment with AVI-4206 did not reduce viral replication in Vero-TMPRSS2 or A549-ACE2[h] cells (*Figure 3A, B*), consistent with previous studies showing that Mac1-deficient SARS-CoV-2 can replicate efficiently in several cell lines (*Taha et al., 2023b*; *Alhammad et al., 2023*). This result stands in contrast to a SARS-CoV-2 protease inhibitor, nirmaltrevir, which potently inhibited replication in both cell lines (EC$_{50}$ 275 and 9.4 nM, respectively) (*Figure 3A, B*). Nonetheless, this experiment, together with a viability assay, indicated no direct cytotoxicity of AVI-4206 at concentrations as high as 100 µM (*Figure 3— figure supplement 1A, B*). Next, we explored whether interferon pretreatment could potentiate the response of AVI-4206 using SARS-CoV-2 replicons (*Taha et al., 2023a*) as we previously demonstrated for a Mac1-deficient SARS-CoV-2 replicon (WA1 N40D mutant, often referred to as N1062D in full length Nsp3) (*Taha et al., 2023b*; *Alhammad et al., 2023*; *Figure 3C*). We did not observe a reduction in viral RNA replication of the Mac1-deficient replicon compared with the wild-type replicon in Vero cells stably expressing ACE2 and TMPRSS2 (VAT) or A549-ACE2[h] cells treated with or without AVI-4206 and 1000 IU/ml of IFN-gamma (*Figure 3—figure supplement 1C, D*). However, there was a modest dose-dependent decrease in replication of the wild-type, but not Mac1-deficient, replicon in A549-ACE2[h] cells (*Figure 3—figure supplement 1D*). When the IFN-gamma dose was increased to 10,000 IU/ml, we observed a small (~1.6-fold) effect for the Mac1-deficient replicon relative to the wild-type replicon (*Figure 3D*). Treatment with the highest dose (100 µM) of AVI-4206 led to a statistically significant, but small (~1.7-fold), reduction in replication for the wild-type, but not Mac1-deficient, replicon (*Figure 3D*). From these experiments, we conclude that cellular models of SARS-CoV-2 infection give, at best, only a narrow window for assessing the efficacy of Mac1 inhibition and that high concentrations of AVI-4206 can achieve a limited anti-viral response without cytotoxicity in an IFN- and Mac1 catalytic activity-dependent manner.

Although the IFN response was not sufficient to control viral replication, it is possible that the changes in ADP-ribosylation, in particular marks catalyzed by PARP14, downstream of IFN treatment could serve as a marker for Mac1 efficacy (*Ribeiro et al., 2025*). To investigate whether downstream signals from PARP14 were specifically erased by Mac1, we used an immunofluorescence assay that showed that Mac1 could remove IFN-γ-induced ADP-ribosylation that is mediated by PARP14 (*Kar et al., 2024*). We stably expressed either wild-type Mac1 or the N40D mutant Mac1 in A549 cells. The data showed that Mac1 expression decreased IFN-γ-induced ADP-ribosylation, whereas the Mac1-N40D mutant did not (*Figure 3E, F*), indicating that Mac1 mediates the hydrolysis of IFN-γ-induced ADP-ribosylation. The PARP14 inhibitor RBN012759 completely blocked IFN-γ-induced ADP-ribosylation (*Figure 3E, F*), further confirming that IFN-γ-induced ADP-ribosylation is mediated by PARP14. AVI-4206 reversed the Mac1-induced hydrolysis of ADP-ribosylation and enhanced the ADP-ribosylation signal in Mac1-overexpressing cells (*Figure 3E, F*), further demonstrating its ability to inhibit the hydrolase activity of Mac1. We further validated this result using different ADP-ribosylation antibodies for immunofluorescence (*Figure 3—figure supplement 2*). However, we observed no strong consistent signals of global pan-ADP-ribose (panADPr) or mono-ADP-ribose (monoADPr) accumulation in infected cells treated with AVI-4206 in immunoblot analyses . Collectively, these results provide further evidence that simple cellular models are insufficient to explore the effects of Mac1

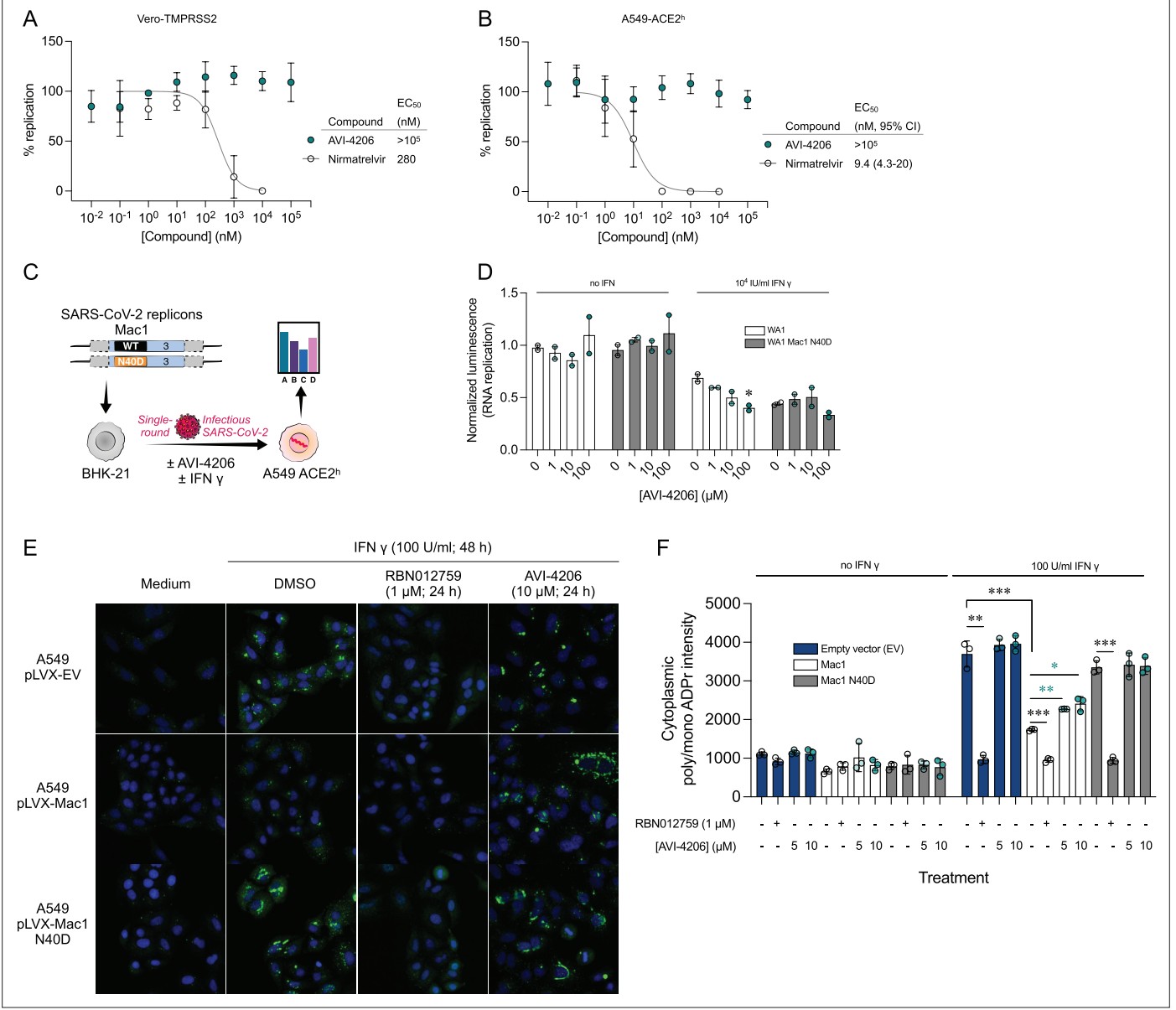

**Figure 3.** Vero-TMPRSS2 (**A**) or A549-ACE2$^h$ (**B**) cells were pretreated with compounds and infected with mNeonGreen reporter SARS-CoV-2. mNeonGreen expression was measured by the Incucyte system. Graphs represent mean ± SD of % replication normalized to the DMSO control 24 post-infection of three independent experiments performed in triplicate. Data were fitted with a sigmoidal dose–response equation using non-linear regression (gray line) and the EC$_{50}$ values are quoted with 95% confidence intervals. (**C**) Schematic of the replicon assay to test the efficacy of AVI-4206 in A549 ACE2$^h$ cells. (**D**) Luciferase readout of A549 ACE2$^h$ cells infected with WA1 or WA1 Mac1 N40D replicons and treated with or without AVI-4206 and IFN-γ at indicated concentrations; *p < 0.05 by two-tailed Student's t-test relative to the no AVI-4206 and no IFN-γ control. Results are plotted as normalized mean ± standard deviation luciferase values of a representative biological experiment containing two technical replicates. (**E**) Representative images of A549 cells stably expressing Mac1 and Mac1-N40D treated with IFN-γ and/or RBN012759 or AVI-4206. DMSO-treated cells are shown as vehicle control. Poly/mono ADPr signal comes from Poly/Mono-ADP Ribose (D9P7Z) Rabbit mAb (CST, 89190S) staining. (**F**) Relative mean cytoplasmic poly/mono ADPr intensity of cells from (**F**). Data shown as mean values ± SD. At least 8000 cells were analyzed in each group, from triplicate wells. Two-tailed Student's t-test was used to compare ADPr intensity levels of each treatment. *p < 0.05; **p < 0.01; ***p < 0.001.

The online version of this article includes the following source data and figure supplement(s) for figure 3:

**Figure supplement 1.** AVI-4206 has limited antiviral efficacy and no cytotoxicity in cellular models of infection.

**Figure supplement 2.** Relative mean cytoplasmic poly/mono ADPr intensity of A549 cells stably expressing Mac1 and Mac1-N40D treated with IFN-γ and/or AVI-4206.

**Figure supplement 3.** ADP-ribosylation profiling during infection by Western Blot.

*Figure 3 continued on next page*

*Figure 3 continued*

**Figure supplement 3—source data 1.** Annotated Iimmunoblot showing pan-ADP-ribose (panADPr) and mono-ADP-ribose (monoADPr) levels in Calu-3 cells under indicated infection conditions.

**Figure supplement 3—source data 2.** Full gels of immunoblot showing pan-ADP-ribose (panADPr) and mono-ADP-ribose (monoADPr) levels in Calu-3 cells under indicated infection conditions.

inhibition and that monitoring specific PARP14-mediated ADP-ribosylation patterns can provide an accessible biomarker for the efficacy of Mac1 inhibition.

## AVI-4206 displays efficacy in organoids and primary cell models

Next, we tested AVI-4206 in a system that more closely replicates both the structural and functional characteristics of the human airway epithelium. We used human airway organoids (HAOs), which are derived from primary stem cells generated from human lungs and grow as complex three-dimensional structures (*Sachs et al., 2019*). These cells can be differentiated into the various cell types found in the airway epithelium, including ciliated, goblet, and basal cells (*Li et al., 2023a*; *Simoneau et al., 2024*). We (*Simoneau et al., 2024*) and others (*Li et al., 2023b*) have utilized differentiated HAOs as a more relevant infection model that encompasses more robust innate immune functions. We therefore sought to test the efficacy of AVI-4206 in HAOs infected with SARS-CoV-2 (*Figure 4A*). The Mac1-deficient virus (WA1 N40D mutant) showed no reduction, 10-fold reduction, and 1000-fold reduction in viral particle production at 24, 48, and 72 hr post-infection compared to the wild-type virus (*Figure 4A*). AVI-4206 treatment reduced viral particle production 10- and 100-fold at 48 and 72 hr post-infection, respectively, and 20 µM AVI-4206 reduced viral particle production by 10-fold at 72 hr post-infection (*Figure 4A*). As we have observed previously (*Taha et al., 2023b*; *Alhammad et al., 2023*), the faster clearance of infection in AVI-4206 treated HAOs, similar to that seen with the

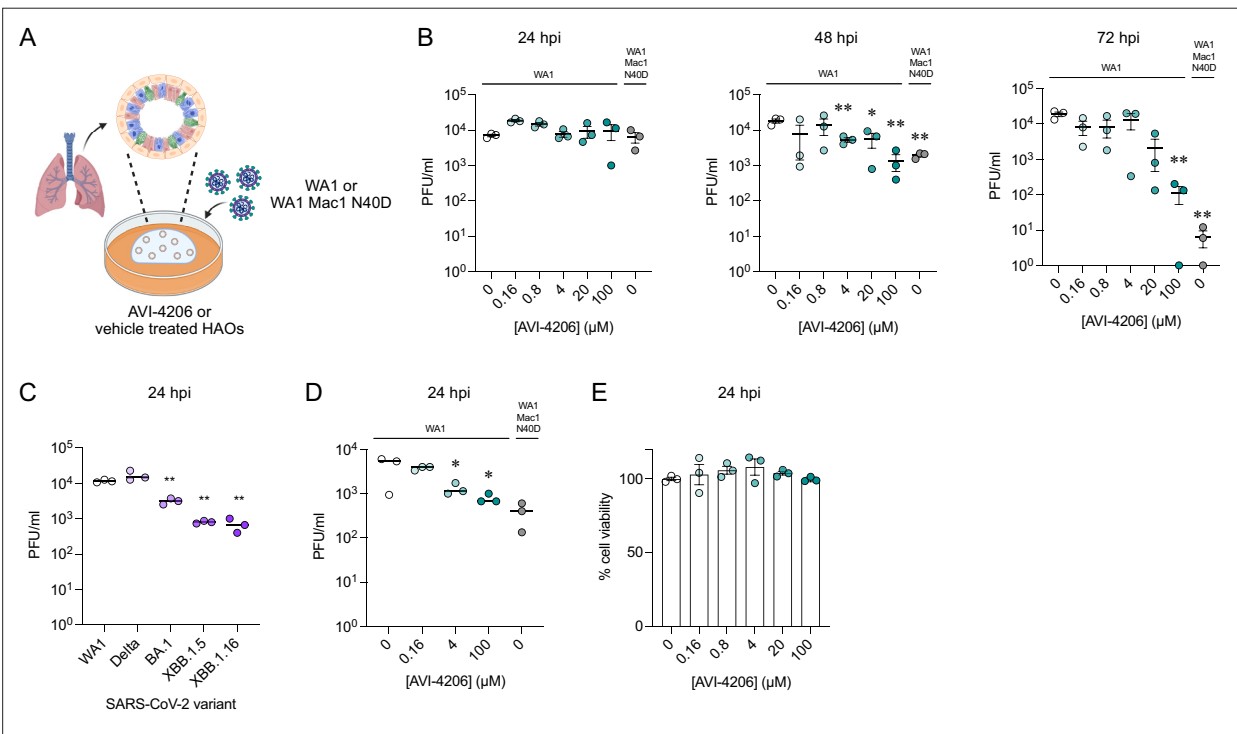

**Figure 4.** AVI-4206 displays efficacy in organoids and primary cell models. (**A**) Schematic of the human airway organoid (HAO) experiment. (**B**) Viral particle production was measured by plaque assay at indicated time points and AVI-4206 concentrations. Error bars indicate SEM. **p < 0.01; *p < 0.05 by two-tailed Student's *t*-test relative to the vehicle control. (**C**) Monocyte-derived macrophages (MDMs) were exposed to SARS-CoV-2, and virus particle production was assessed 24 hr later using a plaque assay. **p < 0.01 by two-tailed Student's t-test relative to WA1. (**D**) Plaque assay of MDMs infected with WA1 or WA1 Mac1 N40D viruses and treated with AVI-4206 at indicated concentrations and IFN-γ at 50 ng/ml. *p < 0.05 by two-tailed Student's *t*-test compared to the untreated control. (**E**) MDMs were incubated with AVI-4206 for 24 hr, after which cytotoxicity was assessed using an ATP-based cytotoxicity assay.

Mac1-deficient virus, is likely due to a potent innate immune response rather than a direct effect of Mac1 on viral replication.

The signs of efficacy in organoids suggested that a more native cellular model might serve as a better surrogate for pharmacological interventions in vivo. We turned to peripheral blood monocyte-derived macrophages (MDMs), key effector cells of the innate immune system with a high capacity to produce and respond to type I interferons via pattern recognition receptors (*Ivashkiv, 2013*; *Kawai and Akira, 2010*). Additionally, due to their interferon-competent nature, primary macrophages have previously been used to understand the role of host PARPs and virally encoded macrodomains in a pathogenic murine coronavirus (*Grunewald et al., 2019*). To validate the potential for MDMs to assess the antiviral activity of AVI-4206, MDMs were first exposed to SARS-CoV-2 variants at a multiplicity of infection (MOI) of 1, and viral particle production was confirmed by plaque assay 24 hr post-infection (*Figure 4C*). To evaluate the efficacy of AVI-4206, MDMs were pretreated with the compound in the presence of IFN-γ, followed by infection with the SARS-CoV-2 WA1 strain. AVI-4206 suppressed viral particle production in a concentration-dependent manner, reducing viral particle production by approximately threefold at 4 μM and fivefold at 100 μM after 24 hr (*Figure 4D*). No statistically significant difference was observed between the antiviral effect of the WA1 N40D mutant and treatment with 100 μM AVI-4206. Importantly, no apparent cytotoxicity was detected at any of the tested concentrations (*Figure 4E*). These findings demonstrate that MDMs provide a physiologically relevant platform for evaluating the antiviral efficacy of macrodomain pharmacological inhibition. While these cellular and organoid experiments gave some indication of an effect of AVI-4206, testing in animal models was required to establish whether this compound had significant activity in reducing viral pathogenesis.

## AVI-4206 has favorable pharmacological properties

Prior to testing the efficacy of AVI-4206 in animal models, we assessed the pharmacological properties of the compound to predict a dosing regime that would provide sufficient target coverage to test efficacy. In parallel with optimizing compounds for potent inhibition of Mac1, as described above, we employed data from standard in vitro assays of metabolism, permeability, and physicochemical properties to drive our medicinal chemistry campaign. Thus, the series leading to AVI-4206 was optimized for stability in mouse liver microsomes and human hepatocytes, low plasma protein binding (good free fraction), and high aqueous solubility (*Figure 5A*). However, the introduction of the urea functionality in this series negatively impacted permeability in Caco-2 monolayers when compared to the parent AVI-219, predicting low oral bioavailability. Indeed, in mouse pharmacokinetic (PK) studies, AVI-4206 showed poor oral bioavailability (<4%), while intrinsic clearance was moderate, about 60% of hepatic blood flow (*Supplementary file 2b*). Bioavailability via the intraperitoneal (IP) route, however, was excellent and free drug concentrations ~100-fold above the biochemical $IC_{50}$ were achieved for 8 hr following a single IP dose at 100 mg/kg (*Figure 5B*, *Supplementary file 2b*). In a separate PK experiment employing a 10 mg/kg IP dose, total exposure of AVI-4206 in lung was higher than in plasma at later time points (*Figure 5C*), suggesting its suitability for an in vivo infection model to validate Mac1 as an antiviral target. Moreover, AVI-4206 showed minimal inhibition of common cytochrome P450 (CYP) isoforms (*Figure 5D*) and screening across a broader panel of potential off targets (*Figure 5E*, *Supplementary file 2c*) identified no significant liabilities among major channels, receptors, or enzymes. Overall, the biochemical potency and PK profile of AVI-4206 suggested the likelihood of sustained target engagement in mice with twice-daily doses (BID) of 100 mg/kg by the IP route allowing us to test proof of concept.

## AVI-4206 is effective in a mouse model of SARS-CoV-2 infection

To assess the efficacy of AVI-4206 in vivo, we used the K18-hACE2 mouse model, which mimics severe SARS-CoV-2 infection (*Zheng et al., 2021*). The K18-hACE2 mouse is a stringent model to test efficacy, as previous studies using potent protease inhibitors did not lead to full survival, unless combined with molnupiravir (*Papini et al., 2024*). In our experiment, animals were divided into three groups (wild-type WA1 virus with compound or vehicle treatment, and a Mac1-deficient mutant-infected positive control) with treatment (AVI-4206 at 100 mg/kg intraperitoneally, nirmatrelvir at 300 mg/kg orally, or vehicle control for each drug) initiated 1 day prior to infection (*Figure 6A*). AVI-4206 or nirmatrelvir was administered twice daily until 5 days post-infection, during which the mice were closely monitored

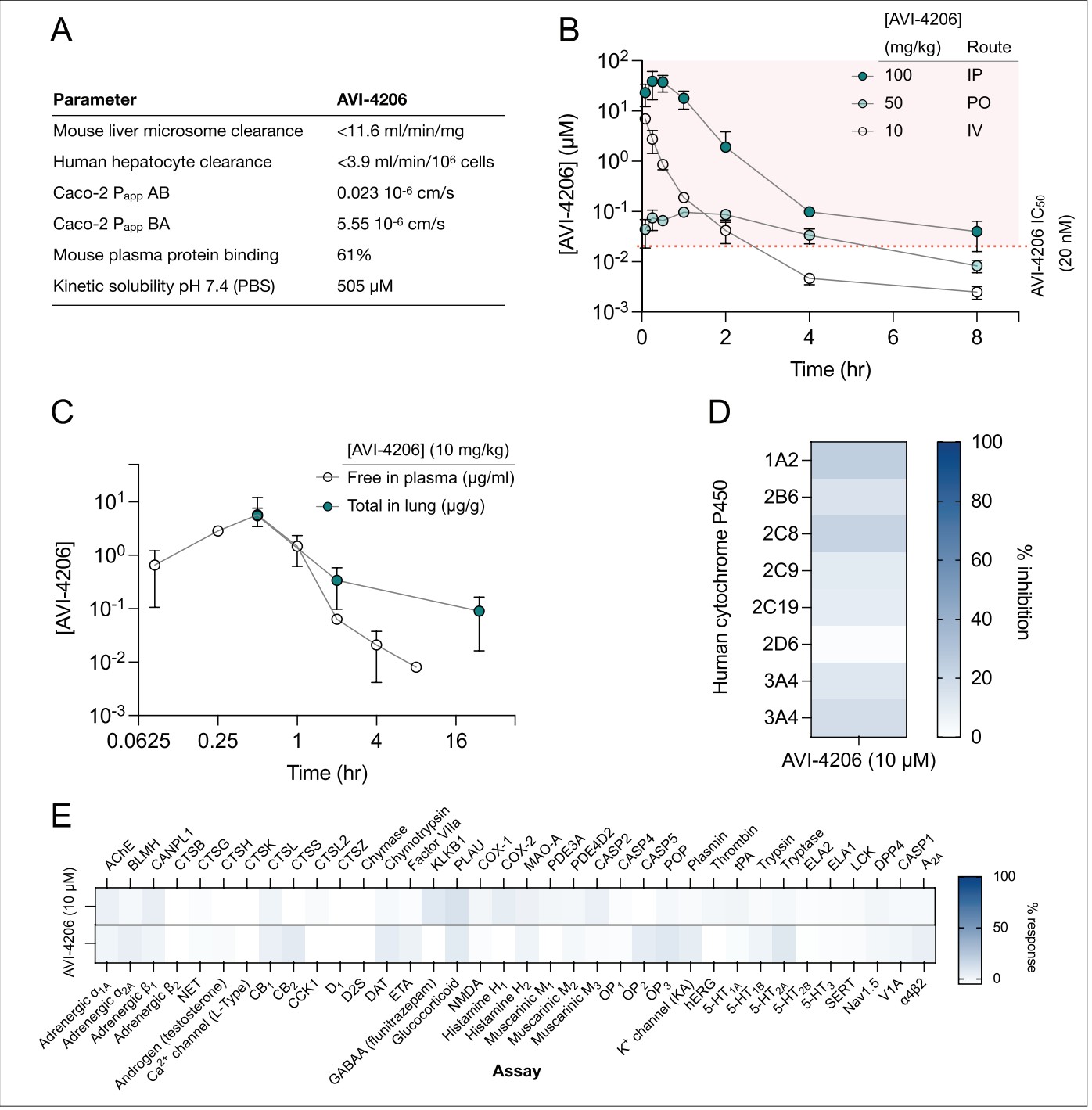

**Figure 5.** AVI-4206 has a favorable pharmacological profile. (**A**) Pharmacokinetic properties of AVI-4206. (**B**) Unbound plasma exposure time course of AVI-4206, corrected for plasma protein binding, following administration by IV, PO, or IP routes in male CD-1 mice at the indicated doses. (**C**) Free plasma exposure of AVI-4206 and total exposure in lung homogenate following an IP dose of 10 mg/kg in female C57BL/6 mice. (**D**) Inhibition of CYP isoforms by AVI-4206 at a fixed concentration of 10 μM. Two experiments were performed with CYP3A4 using different positive controls. (**E**) Heatmap of AVI-4206 activity in an off-target safety panel including receptors, ion channels, and proteases, showing no antagonist response >15% at 10 μM.

for disease parameters such as weight loss and hunched posture. Both vehicle-treated groups experienced weight loss starting at 3–4 days post-infection and continued losing weight until the end of the study at 7 days post-infection (*Figure 6B, C*). The AVI-4206 and nirmatrelvir treated groups experienced weight loss starting at 4 days post-infection, but the extent of weight loss was about

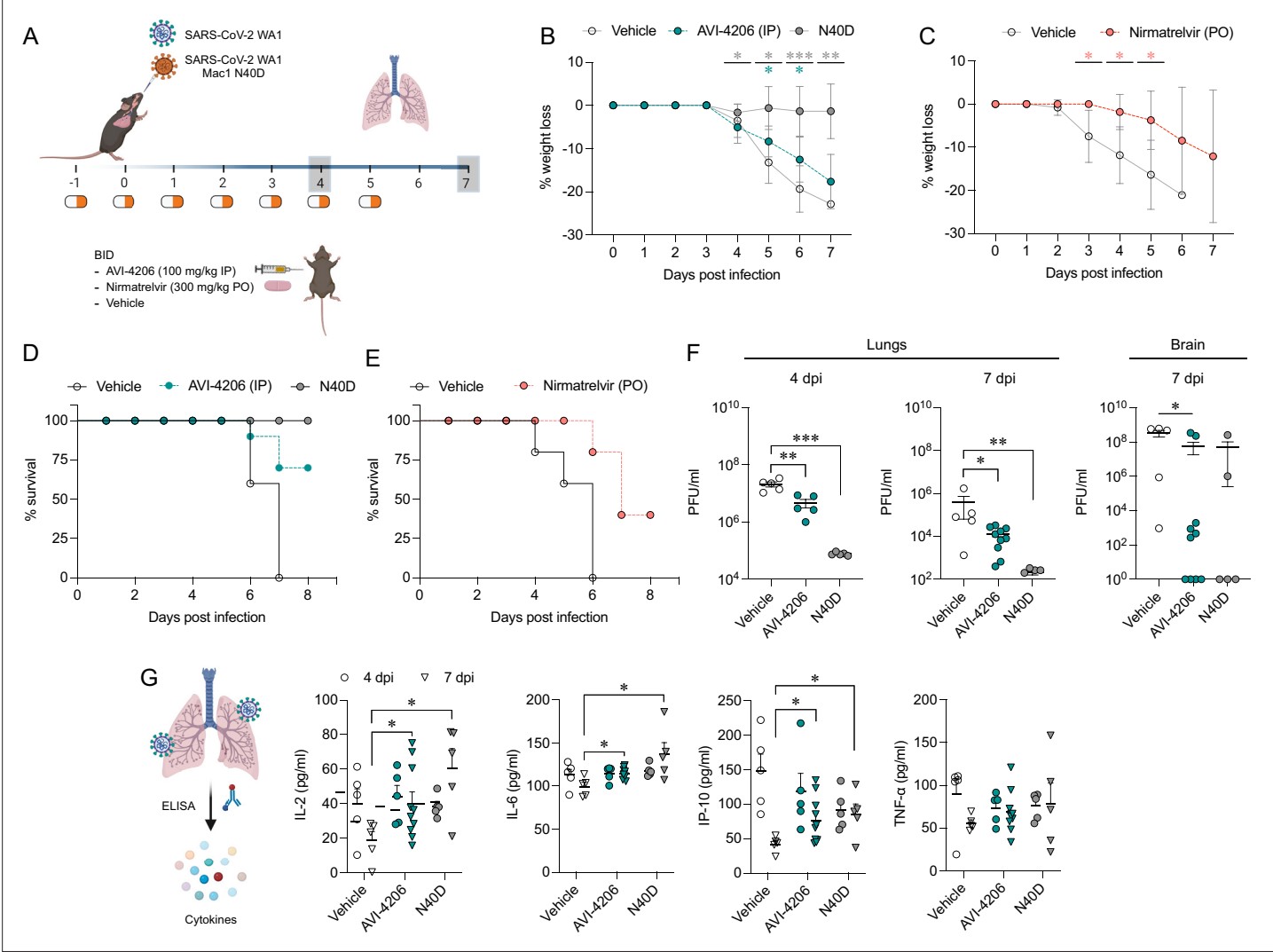

**Figure 6.** AVI-4206 reduces viral replication and increases survival and cytokine abundance in vivo. (**A**) K18-hACE2 mice were intranasally infected and dosed as indicated with either AVI-4206 (*n* = 15, intraperitoneally), nirmatrelvir (*n* = 5, per os), or vehicle (*n* = 10 for the AVI-4206 group or *n* = 5 for the nirmatrelvir group). Mice infected with the WA1 N40D mutant, which lacks Mac1 catalytic activity, served as a positive control (*n* = 10). Lungs were harvested at indicated time points for virus titration by plaque assay. (**B**) The percent body weight loss for all animals treated with AVI-4206 (100 mg/kg IP) (**C**) or nirmatrelvir (300 mg/kg PO). The data are presented as mean ± SD. *p < 0.05; **p < 0.01; ***p < 0.001 by two-tailed Student's *t*-test relative to the vehicle control at each time point. (**D**) Survival curve plotted based on the percent weight loss humane endpoint (20%) for AVI-4206 and (**E**) nirmatrelvir. (**F**) Viral load in the lungs and brain of infected mice at the indicated time points. The data are shown as mean ± SEM. *p < 0.05; **p < 0.01 by Mann–Whitney's test relative to the vehicle control. (**G**) Schematics and graphs demonstrating the abundance of indicated cytokines at 4 and 7 days post-infection in the lungs of infected mice. The data are presented as mean ± SEM. *p < 0.05; **p < 0.01 by two-tailed Student's *t*-test relative to the vehicle control at each time point. None of the mice reached the humane endpoint at day 4 post-infection. For mice that reached the humane endpoint before day 7 post-infection, the tissues were collected and analyzed with mice at the 7-day time point.

The online version of this article includes the following figure supplement(s) for figure 6:

**Figure supplement 1.** Lung histology.

**Figure supplement 2.** Lower dose AVI-4206 reduces viral replication and increases survival in vivo.

**Figure supplement 3.** AVI-4206 suppresses replication of mouse-adapted SARS-CoV-2 in wild-type mice.

5–10% lower on average at days 5–7 post-infection compared with their respective vehicle-treated groups (*Figure 6B, C*). Consequently, ~70% of the AVI-4206 treated group and 40% of the nirmatrelvir treated group survived, whereas all mice in the vehicle treated groups died by the end of the study based on the humane endpoints of hunched posture or >20% decrease in body weight (*Figure 6D, E*). Consistent with previous studies (*Taha et al., 2023b*), none of the mice in the mutant-infected positive

control group experienced weight loss, and all survived the infection (*Figure 6D*). These results indicate that AVI-4206 can significantly reduce disease severity and prevent death in the K18-hACE2 model and is comparable to the FDA-approved main protease inhibitor nirmatrelvir.

To understand the mechanism of AVI-4206 action during the course of infection, mice from each group were euthanized at either day 4 or 7. We observed that AVI-4206 treatment reduced viral load in the lungs by ~10- and ~100-fold at 4 and 7 days post-infection, respectively, and reduced transmission to the brain (*Figure 6F*). Although it is possible that AVI-4206 crosses the blood–brain barrier, it is more likely that the reduction of viral load in the brain is as a consequence of a reduction in overall systemic viral load. The prevention of virus localization to the brain is especially important in this model because human ACE2 overexpression allows virus replication and spread to brain tissue, which ultimately leads to encephalitis and the death of infected mice (*Bao et al., 2020*; *Oladunni et al., 2020*). The faster clearance of viral load in the lungs for AVI-4206 treated and Mac1-deficient virus infected mice compared with the vehicle-treated mice, rather than an early antiviral effect post-infection, is consistent with an immune response mediated mechanism rather than a direct antiviral mechanism.

To further investigate the antiviral mechanism of AVI-4206, we measured the abundance of the antiviral cytokines IP-10, IL-2, IL-6, and TNF-α in lung tissue at 4 and 7 days post-infection (*Figure 6G*). We found that levels of all of these cytokines were elevated at 4 days post-infection. At 7 days post-infection, the AVI-4206 treated and Mac1-deficient virus infected mice maintained significantly higher levels of IP-10, IL-2, and IL-6 ($p < 0.05$) compared to the vehicle treated group; TNF-α showed a similar trend but did not reach statistical significance (*Figure 6G*). The lower levels of cytokines in the vehicle-treated group at 7 days post-infection are likely mediated by the immune-suppressive capability of SARS-CoV-2 macrodomain. However, when the macrodomain is inactivated, either through AVI-4206 treatment or infection with Mac-1 defective variant, the antiviral response is enhanced, which blocks viral replication. The cumulative cytokine abundance (IP-10, IL-2, and IL-6) indicates an antiviral immune response, likely mediated through the activation of the NF-κB pathway (*Robertson et al., 2023*; *Neufeldt et al., 2022*). Caspase 3 staining shows that AVI-4206 treatment reduces apoptosis in the lungs compared to vehicle controls. Additionally, histology reveals a reduction in Caspase 3 staining and Masson's Trichrome staining for collagen deposition, which are surrogates for lung pathology, in the lungs of AVI-4206 treated animals (*Figure 6—figure supplement 1*). We additionally tested the efficacy of AVI-4206 at a lower dosage of 30 mg/kg using the same experimental setup. Even at this lower dose, AVI-4206 enhanced survival and produced lower viral load at 7 days post-infection relative to vehicle (*Figure 6—figure supplement 2*), but to a more modest degree than at the higher dose (*Figure 6*). Finally, we assessed the antiviral efficacy of AVI-4206 using a wild-type mouse model infected with a mouse-adapted SARS-CoV-2 strain, which exhibits reduced neuroinvasiveness compared to ACE2 transgenic models. Treatment with AVI-4206 resulted in a measurable reduction in pulmonary viral load as early as 2 days post-infection (*Figure 6—figure supplement 3*). Collectively, our observations of enhanced survival of mice in a lethal model, reduced viral load in multiple mouse models, and the observed increase in antiviral cytokines suggest that AVI-4206 is capable of potentiating the host immune response, thereby reducing disease severity.

## Discussion

Here, we provide strong pharmacological evidence validating Mac1 and de-ADP-ribosylation as a therapeutic target for SARS-CoV-2. AVI-4206 is a competitive inhibitor that blocks the ADP-ribosylhydrolase activity of Mac1. This activity antagonizes the PARP-mediated ADP-ribosylation that is part of the antiviral interferon response. Although mechanistic links are still emerging between specific post-translational modifications and an effective antiviral response, our pharmacologic studies add to the genetic and biochemical evidence of the importance of this signaling axis for viral replication in vivo (*Taha et al., 2023b*; *Alhammad et al., 2023*). Our work also adds to the growing role of modulating ADP-ribosylation signaling in therapeutic development (*Dasovich and Leung, 2023*). For example, inhibitors of PARP1, which catalyzes the addition of poly- ADP-ribose marks, have been developed for treating tumors with mutations in either *BRCA1* or *BRCA2* (*Lord and Ashworth, 2017*) and inhibitors of PARG, which catalyzes the removal of α(1″–2′) O-glycosidic linkages in PAR chains, are under investigation for a variety of cancers (*Slade, 2020*). The presence of macrodomains and experimental evidence for them as interferon signaling antagonists in other diverse viruses, such as

Chikungunya (*McPherson et al., 2017*), suggests that inhibiting this target class may be effective for treatment of other virally induced diseases beyond COVID.

While AVI-4206 is protective in an animal model of infection, it was initially developed without many of the normal intermediate markers of improvement in cellular models. This discordance was expected based on the mechanism of action, as interferon-based antiviral activity likely requires intra- and inter-cellular and systemic communication between different cell types (*Platanias, 2005*). The limited replication defect difference between wild-type and Mac1-deficient viruses in cellular models renders them largely ineffective as a model to test the effects of macrodomain inhibition, which has led others to question the validity of Mac1 as a target (*Lee et al., 2024*). AVI-4206 did in fact demonstrate modest antiviral activity only in the presence of exogenous IFN in cells, which is consistent with most other studies that have examined Mac1 activity (*Taha et al., 2023b*; *Alhammad et al., 2023*; *Kerr et al., 2024*). While a larger replication defect is observed in HAOs likely due to their more relevant antiviral innate immune responses (*Simoneau et al., 2024*), the highest dose of AVI-4206 does not achieve the magnitude of the replication defect of the Mac1-deficient virus. This may reflect an unoptimized prophylactic dosing schedule or the need to better tune the pharmacological properties of the inhibitor. Taken together, the concordance of in vitro (HTRF) and cellular target engagement assays (CETSA) stands out as particularly important in the development path of macrodomain inhibitors. Nonetheless, developing a cellular model for assessing macrodomain function and inhibition is important. We have now been able to leverage SARS-CoV-2–infected macrophages, where an active immune response is induced, including inflammasome activation (*Sefik et al., 2022*), to assess the antiviral efficacy of AVI-4206 by detecting and quantifying transient viral particle production following infection. This macrophage-based cell model will enable more physiologically relevant evaluation of macrodomain inhibitors. This cell model will also be important for elucidating the molecular details underlying how macrodomain inhibition results in antiviral activity. In addition to dissecting molecular mechanisms of macrodomain function and inhibition, future efforts will focus on improving PK properties, including a cellular efflux liability that results in low oral bioavailability of AVI-4206.

In conclusion, AVI-4206 blocks the viral enzymatic removal of post-translational modifications important for the immune response, which is an important mechanism for blocking virus replication and reducing disease severity. Notably, Mac1 represents a second pharmacologically validated enzymatic domain within Nsp3: a recently developed inhibitor of the papain-like protease (PLpro) domain also shows efficacy in animal models and acts by removing a distinct set of host post-translational modifications (ubiquitin and interferon-stimulated gene 15 (ISG15)) (*Tan et al., 2024*). In summary, AVI-4206 provides proof of concept for the validation of a novel antiviral target, Mac1. By restoring the antiviral immune response, the novel mechanism of action of AVI-4206 could be synergistic or additive with orthogonally acting direct antivirals, such as protease and polymerase inhibitors, in combination therapies for the treatment of SARS-CoV-2 infection and beyond.

## Methods

### Synthetic chemistry

#### General experimental procedures

Unless otherwise noted, all chemical reagents and solvents used are commercially available. AVI-92 and AVI-219 were synthesized as previously described (*Gahbauer et al., 2023*). Reverse phase chromatography was carried out on one of the following instruments: (1) Waters 2535 Separation module with Waters 2998 Photodiode Array Detector. Separations were carried out on XBridge Preparative C18, 19 × 50 mm column at ambient temperature using a mobile phase gradient of water–acetonitrile–0.1% formic acid. (2) Gilson GX-281 instrument, separations using Xtimate Prep C18, 21.2*250 mm, 150 Å, 10 μm particle size column. (3) Agilent 1260 Infinity systems equipped with DAD and mass detector. Separations carried out on Chromatorex 18 SMB100-5T 100 × 19 mm 5 μm column using mobile phase gradient of water/methanol/0.005% HCl. Chiral separations were carried out on CHIRALPAK IA (250 × 21 mm, 5 mkm)-II column at ambient temperature using a mobile phase of hexane (0.3% DEA): IPA:MeOH, 90:5:5. LC/MS data were acquired by one of the following instruments: (1) Waters Acquity UPLC QDa mass spectrometer equipped with Quaternary Solvent Manager, Photodiode Array Detector, and Evaporative Light Scattering Detector. Separations were carried out with Acquity UPLC BEH C18 1.7 mm, 2.1 × 50 mm column at 25°C, using a mobile phase

gradient of water–acetonitrile containing a constant 0.1% formic acid. Detection: UV (254 nm), ELS, and MS (ESI, positive mode), (2) Agilent 1100 Series LC/MSD system with DAD/ELSD Alltech 2000ES and Agilent LC/MSD VL (G1956B), SL (G1956B) mass-spectrometer, (3) Agilent 1200 Series LC/MSD system with DAD/ELSD Alltech 3300 and Agilent LC/MSD G6130A, G6120B mass-spectrometer, (4) Agilent Technologies 1260 Infinity LC/MSD system with DAD/ELSD Alltech 3300 and Agilent LC/MSD G6120B mass-spectrometer, or (5) Agilent Technologies 1260 Infinity II LC/MSD system with DAD/ELSD G7102A 1290 Infinity II and Agilent LC/MSD G6120B mass-spectrometer. Separations were carried out with InfinityLab Poroshell 120 SB-C18 4.6 × 30 mm 2.7 µm column at 25°C, using a mobile phase gradient of water–acetonitrile containing a constant 0.1% formic acid. Detection using DAD1A 215 nm, DAD1B 254 nm MSD – single quadrupole, AP-ESI (positive/negative mode switching). (6) Agilent 1200 Infinity LC with an Agilent 1956 single quadrupole MS using electrospray ionization. Separations were carried out on a SunFire C18 (4.6 × 50 mm, 3.5 µm) column at 50°C using a mobile phase gradient of water (10 mmol NH$_4$HCO$_3$)/acetonitrile. Detection: UV (214, 254 nm) and MS (ESI, POS mode, 103–100 atomic mass units). Chemical shifts are reported in units of ppm. NMR spectra were referenced relative to residual NMR solvent peaks. Coupling constants ($J$) are reported in hertz (Hz). NMR spectra were recorded on one of the following instruments: (1) Bruker AVANCE DRX 500 (500 MHz magnet with 5 mm QNP $^{31}$P/$^{13}$C/$^{15}$N and 5 mm TXI probe), (2) Agilent ProPulse 600 (600 MHz magnet with 5 mm OneNMR probe), and (3) Bruker Avance III HD 400 MHz spectrometer.

## 1-((8-Amino-9*H*-pyrimido[4,5-b]indol-4-yl)amino)pyrrolidin-2-one

**Chemical structure 1.** 1-((8-Amino-9H-pyrimido[4,5-b]indol-4-yl)amino)pyrrolidin-2-one.

A solution 3-fluoro-2-nitroaniline (11 g, 70.51 mmol) in acetic anhydride (20 ml) was stirred at room temperature for 16 hr. The reaction mixture was filtered and the solids were washed with petroleum ether (100 ml) and dried to obtain 10.7 g (77%) of *N*-(3-fluoro-2-nitrophenyl)acetamide as a brown solid. LC–MS (ESI): *m/z* = 199.3 (M+H)$^+$.

To a solution of *N*-(3-fluoro-2-nitrophenyl)acetamide (10.7 g, 54.04 mmol) in DMF (100 ml) was added methyl 2-isocyanoacetate (8.02 g, 81.06 mmol) and potassium carbonate (14.92 g, 108.08 mmol). After stirring at 80°C for 2 hr, the reaction mixture was cooled to room temperature, acidified with 2 N HCl (ca. 2000 ml), and extracted with ethyl acetate (300 ml *3). The combined organic layers were washed with brine (100 ml), dried over sodium sulfate, and concentrated under reduced pressure. The residue was purified by silica gel chromatography (10:1 petroleum ether/ethyl acetate) to obtain 11 g (73%) of methyl 2-(3-acetamido-2-nitrophenyl)-2-isocyanoacetate as a yellow solid. LC–MS (ESI): *m/z* = 278.2 (M+H)$^+$.

To a solution of methyl 2-(3-acetamido-2-nitrophenyl)-2-isocyanoacetate (11 g, 39.71 mmol) in *glacial* acetic acid (100 ml), was added slowly zinc dust (25.81 g, 397.10 mmol) in two portions. After

stirring at 60°C for 2 hr, the reaction mixture was cooled to room temperature, filtered, and washed with THF. The filtrate was concentrated under reduced pressure and purified by silica gel chromatography (10:1 dichloromethane/methanol) to obtain 6.2 g (63%) of methyl 7-acetamido-2-amino-1*H*-indole-3-carboxylate as a yellow solid. LC–MS (ESI): *m/z* = 248.3 (M+H)$^+$.

A solution of methyl 7-acetamido-2-amino-1*H*-indole-3-carboxylate (6.2 g, 25.10 mmol) in formamide (450 ml) was stirred at 220°C for 2 hr. The reaction mixture was then cooled to room temperature and poured into 100 ml of water. The resulting mixture was allowed to stand for 15 min before the solids were collected by filtration, washed with water, and dried to obtain 4.1 g of a 1:2 mixture of *N*-(4-hydroxy-9*H*-pyrimido[4,5-b]indol-8-yl)acetamide and *N*-(4-hydroxy-9*H*-pyrimido[4,5-b]indol-8-yl) formamide. This mixture was taken in methanol (25 ml) and aqueous 12 N NaOH (25 ml). After stirring at 60°C for 16 hr, the reaction mixture was then cooled to room temperature, concentrated under reduced pressure to remove methanol, and the residue was poured into 100 ml of water. The resulting mixture was allowed to stand for 15 min before the solids were collected by filtration, washed with water, and dried to obtain 3.5 g (70%) of 8-amino-9*H*-pyrimido[4,5-b]indol-4-ol as a brown solid. LC–MS (ESI): *m/z*=201.2 (M+H)$^+$.

A solution of 8-amino-9*H*-pyrimido[4,5-b]indol-4-ol (3.5 g, 17.5 mmol) in formamide (30 ml) was stirred at 150°C. After 6 hr, the reaction mixture was cooled to room temperature and poured into water (200 ml). The resulting mixture was allowed to stand for 15 min before the solids were collected by filtration, washed with water, and dried to obtain 3.5 g (88%) of *N*-(4-hydroxy-9*H*-pyrimido[4,5-b] indol-8-yl)formamide as a brown solid. LC–MS (ESI): *m/z* = 229.2 (M+H)$^+$.

To a solution of *N*-(4-hydroxy-9*H*-pyrimido[4,5-b]indol-8-yl)formamide (3.5 g, 15.35 mmol) in phosphorous oxychloride (30 ml) was added *N*,*N*-diisopropylethylamine (5.94 g, 46.05 mmol). After refluxing for 16 hr, the reaction mixture was cooled to room temperature, concentrated, and poured into water (20 ml). The resulting solid was filtered to obtain 500 mg of a mixture of *N*-(4-chloro-9*H*-pyrimido[4,5-b]indol-8-yl)formamide and 4-chloro-9*H*-pyrimido[4,5-b]indol-8-amine as a black solid. This mixture was taken in 4 N HCl in dioxane (15 ml). After stirring at room temperature for 4 hr, the reaction mixture was concentrated under reduced pressure, the residue was adjusted to pH 7 with aqueous Na$_2$CO$_3$, and extracted with ethyl acetate (3 × 30 ml). The organic layers were dried over sodium sulfate, concentrated under reduced pressure, and the residue was purified by reverse phase chromatography (water/acetonitrile/0.1% ammonium bicarbonate) to obtain 320 mg (10%) of 4-chloro-9*H*-pyrimido[4,5-b]indol-8-amine as a white solid. $^1$H NMR (500 MHz, DMSO) δ 12.42 (s, 1H), 8.74 (s, 1H), 7.58 (d, *J* = 7.8 Hz, 1H), 7.25–7.08 (m, 1H), 6.93 (d, *J* = 7.7 Hz, 1H), 5.76 (s, 2H). LC–MS (ESI): *m/z* = 219.2 (M+H)$^+$.

A mixture of 4-chloro-9*H*-pyrimido[4,5-b]indol-8-amine (28 mg, 0.13 mmol) and 1-aminopyrrolidin-2-one hydrochloride (35 mg, 0.26 mmol) in isopropanol/water (10: 1, 1.1 ml) was heated to 100°C for 18 hr. The reaction mixture was filtered, the residue was washed with ethyl acetate and dried to obtain 28 mg (77%) of 1-((8-amino-9*H*-pyrimido[4,5-b]indol-4-yl)amino)pyrrolidin-2-one as brown solid. $^1$H NMR (DMSO-d$_6$, 400 MHz) δ 12.99 (br s, 1H), 8.62 (s, 1H), 7.92 (br d, 1H, *J* = 7.5 Hz), 7.27 (t, 1H, *J* = 7.9 Hz), 7.05 (br d, 1H, *J* = 7.5 Hz), 3.70 (br t, 2H, *J* = 6.9 Hz), 2.44–2.53 (m, 2H), 2.20 (br t, 2H, *J* = 7.4 Hz). $^{13}$C NMR (METHANOL-d$_4$, 100 MHz) δ 175.9, 155.9, 154.3, 153.2, 132.5, 125.7, 121.9, 119.4, 111.3, 111.1, 97.0, 48.6, 28.5, 15.9. LC–MS (ESI): *m/z* = 283 (M+H)$^+$.

## 1-Amino-5,5-dimethylpyrrolidin-2-one hydrochloride

**Chemical structure 2.** 1-Amino-5,5-dimethylpyrrolidin-2-one hydrochloride.

To a cooled (0°C) solution of 5,5-dimethylpyrrolidin-2-one (3 g, 26.54 mmol) in THF (60 ml) was added sodium hydride (2.13 g, 53.09 mmol), followed by addition of (aminooxy)diphenylphosphine oxide (12.4 g, 53.09 mmol) after 30 min. After stirring the resultant white suspension at 0°C for 2 hr,

the reaction mixture was filtered through a Celite pad, the filtrate was concentrated and purified by silica gel chromatography (10:1 dichloromethane/methanol) to afford 3 g (75%) of 1-amino-5,5-dimethylpyrrolidin-2-one as yellow oil. LC–MS (ESI): $m/z$ = 129.1 (M+18)[+].

A solution of 1-amino-5,5-dimethylpyrrolidin-2-one (1.5 g, crude) in 4 N HCl in dioxane (15 ml) was stirred at room temperature for 4 hr. The mixture was concentrated under reduced pressure, residue was triturated with diethyl ether and filtered to afford 1 g (53%) of 1-amino-5,5-dimethylpyrrolidin-2-one hydrochloride salt as a white solid. [1]H NMR (500 MHz, DMSO) δ 9.48 (s, 3 H), 2.39 (t, 2H, $J$ = 7.8 Hz), 1.90 (t, 2H, $J$ = 7.8 Hz), 1.30 (s, 6H). LC–MS (ESI): $m/z$ = 129.1 (M+18)[+].

## AVI-4051

**Chemical structure 3.** AVI-4051.

To a solution of 1-((8-amino-9$H$-pyrimido[4,5-b]indol-4-yl)amino)pyrrolidin-2-one (20 mg, 0.071 mmol) and triethylamine (0.040 ml, 0.28 mmol) in THF (1 ml), was added cyclopropyl isocyanate (24 mg, 0.28 mmol). After stirring at 65°C for 48 hr, the reaction mixture was purified by reverse phase chromatography (water/acetonitrile/0.1% formic acid) to obtain 12 mg (41%) of 1-cyclopropyl-3-(4-((2-oxopyrrolidin-1-yl)amino)-9$H$-pyrimido[4,5-b]indol-8-yl)urea formic acid salt (AVI-4051) as a white solid. [1]H NMR (DMSO-d$_6$, 400 MHz) δ 11.81 (br s, 1H), 9.31 (s, 1H), 8.50 (br s, 1H), 8.41 (s, 1H), 7.99 (d, 1H, $J$ = 7.8 Hz), 7.63 (d, 1H, $J$ = 7.8 Hz), 7.21 (t, 1H, $J$ = 7.9 Hz), 6.74 (br s, 1H), 3.70 (br t, 2H, $J$ = 7.1 Hz), 3.12–3.17 (m, 2H), 2.54–2.65 (m, 1H), 2.39–2.43 (m, 2H), 0.98 (t, 2H, $J$ = 7.1 Hz), 0.68–0.70 (m, 2H). LC–MS (ESI): $m/z$ = 366 (M+H)[+].

## AVI-1501

**Chemical structure 4.** AVI-1501.

To a solution of 1-((8-amino-9*H*-pyrimido[4,5-b]indol-4-yl)amino)pyrrolidin-2-one (15 mg, 0.053 mmol) and triethylamine (0.015 ml, 0.11 mmol) in THF (1 ml), was added acetyl chloride (0.004 ml, 0.056 mmol). After stirring at 65°C for 3 hr, the reaction mixture was purified by reverse phase chromatography (water/acetonitrile/0.1% formic acid) to obtain 9 mg (50%) of *N*-(4-((2-oxopyrrolidin-1-yl)amino)-9*H*-pyrimido[4,5-b]indol-8-yl)acetamide formic acid (**AVI-1501**) as a white solid. $^1$H NMR (METHANOL-d$_4$, 400 MHz) δ 8.39 (s, 1H), 7.90 (d, 1H, *J* = 7.8 Hz), 7.45 (d, 1H, *J* = 7.8 Hz), 7.19–7.21 (m, 1H), 3.81–3.85 (m, 2H), 2.59–2.63 (m, 2H), 2.27–2.31 (m, 5H). LC–MS (ESI): *m/z* = 325 (M+H)$^+$.

## AVI-3367

**Chemical structure 5.** AVI-3367.

To a solution of 1-((8-amino-9*H*-pyrimido[4,5-b]indol-4-yl)amino)pyrrolidin-2-one (15 mg, 0.053 mmol) and triethylamine (0.015 ml, 0.11 mmol) in THF (1 ml), was added ethyl chloroformate (0.005 ml, 0.056 mmol). After stirring at 65°C for 18 hr, the reaction mixture was purified by reverse phase chromatography (water/acetonitrile/0.1% formic acid) to obtain 2.7 mg (13%) of ethyl (4-((2-oxopyrrolidin-1-yl)amino)-9*H*-pyrimido[4,5-b]indol-8-yl)carbamate formic acid salt (**AVI-3367**) as tan solid. $^1$H NMR (METHANOL-d4, 400 MHz) δ 8.42 (s, 1H), 7.94 (d, 1H, *J* = 7.8 Hz), 7.59 (br s, 1H), 7.28 (t, 1H, *J* = 7.9 Hz), 4.1–4.26–4.30 (m, 2H), 3.84 (t, 2H, *J* = 7.1 Hz), 2.60 (t, 2H, *J* = 8.0 Hz), 2.30–2.33 (m, 2H), 1.36–1.39 (m, 3H). LC–MS (ESI): *m/z* = 355 (M+H)$^+$.

## AVI-1500

**Chemical structure 6.** AVI-1500.

To a solution of 4-chloro-9*H*-pyrimido[4,5-b]indol-8-amine (50 mg, 0.23 mmol) and triethylamine (0.064 ml, 0.46 mmol) in THF (2 ml), was added ethyl isocyanate (0.018 ml, 0.23 mmol). After stirring at

65°C for 18 hr, the reaction mixture was filtered. The residue was washed with ethyl acetate and dried to obtain 50 mg of 1-(4-chloro-9*H*-pyrimido[4,5-b]indol-8-yl)-3-ethylurea as a white solid that was used without further purification. [1]H NMR (DMSO-d$_6$, 400 MHz) δ 12.39 (br s, 1H), 8.80 (s, 1H), 8.43 (s, 1H), 7.96 (d, 1H, *J* = 7.6 Hz), 7.72 (d, 1H, *J* = 7.8 Hz), 7.35 (t, 1H, *J* = 7.9 Hz), 6.38 (s, 1H), 3.18–3.21 (m, 2H), 1.12 (t, 3H, *J* = 7.2 Hz). LC–MS (ESI): *m/z* = 290, 292 (M+H)[+].

A mixture of 1-(4-chloro-9*H*-pyrimido[4,5-b]indol-8-yl)-3-ethylurea (26 mg, 0.09 mmol) and 1-aminopyrrolidin-2-one hydrochloride (25 mg, 0.18 mmol) in isopropanol/water (10: 1, 1.1 ml) was heated to 100°C for 18 hr. The reaction mixture was purified by reverse phase chromatography (water/acetonitrile/0.1% formic acid) to obtain 8 mg (20%) of 1-ethyl-3-(4-((2-oxopyrrolidin-1-yl)amino)-9*H*-pyrimido[4,5-b]indol-8-yl)urea formic acid salt (**AVI-1500**) as a white solid. [1]H NMR (DMSO-d6, 400 MHz) δ 11.75 (br s, 1H), 9.31 (s, 1H), 8.53 (s, 1H), 8.41 (s, 1H), 7.97 (d, 1H, *J* = 7.8 Hz), 7.63 (d, 1H, *J* = 7.8 Hz), 7.20 (t, 1H, *J* = 7.9 Hz), 6.37 (br s, 1H), 3.70 (t, 2H, *J* = 7.1 Hz), 3.17–3.20 (m, 2H), 2.39–2.41 (m, 2H), 2.12–2.16 (m, 2H), 1.09–1.13 (m, 3H). [13]C NMR (DMSO-d$_6$, 100 MHz) δ 173.5, 156.2, 155.9, 155.5, 155.0, 128.5, 125.5, 121.3, 120.3, 117.1, 116.7, 96.4, 48.4, 34.8, 28.9, 16.7, 15.9. LC–MS (ESI): *m/z* = 354 (M+H)[+].

## AVI-3762 and AVI-3763

**Chemical structure 7.** AVI-3762 and AVI-3763.

4-Chloro-7-fluoro-9*H*-pyrimido[4,5-b]indole (123 mg, 0.55 mmol) and 1-amino-5-phenyl-pyrrolidin-2-one (117 mg, 0.66 mmol) in a mixture of dioxane*HCl/IPA (1.5 ml/1.5 ml) was stirred at 95°C overnight. Upon completion, the mixture was cooled to rt and concentrated under reduced pressure. The crude material was purified by HPLC (30–80% MeOH/H$_2$O) to afford 1-((7-fluoro-9*H*-pyrimido[4,5-b]indol-4-yl)amino)-5-phenylpyrrolidin-2-one (69 mg, HCl salt, 34% yield). It was further separated by chiral chromatography (Hexane-IPA-MeOH, 50-25-25) to obtain **AVI-3762** (27 mg, retention time = 14.04 min, 99% optic ee) and **AVI-3763** (26 mg, retention time = 11.17 min, 100% optic ee).

### AVI-3762
[1]H NMR (DMSO-d$_6$, 400 MHz) δ 12.22 (br s, 1H), 9.41 (br s, 1H), 8.44 (s, 1H), 8.26 (dd, 1H, *J* = 5.4, 8.8 Hz), 7.48 (br d, 2H, *J* = 7.3 Hz), 7.22–7.36 (m, 4H), 7.10 (ddd, 1H, *J* = 2.2, 8.8, 9.7 Hz), 5.21 (br s, 1H), 2.55–2.65 (m, 3H), 1.87–1.92 (m, 1H). LC–MS (ESI): *m/z* = 362 (M+H)[+].

### AVI-3763
[1]H NMR (DMSO-d$_6$, 400 MHz) δ 12.22 (s, 1H), 9.41 (br s, 1H), 8.44 (s, 1H), 8.26 (dd, 1H, *J* = 5.4, 8.8 Hz), 7.48 (br d, 2H, *J* = 7.5 Hz), 7.22–7.36 (m, 4H), 7.10 (ddd, 1H, *J* = 2.4, 8.8, 9.7 Hz), 5.21 (br s, 1H), 2.57–2.65 (m, 3H), 1.87–1.92 (m, 1H). LC–MS (ESI): *m/z* = 362 (M+H)[+].

## AVI-3764 and AVI-3765

**Chemical structure 8.** AVI-3764 and AVI-3765.

4-Chloro-7-fluoro-9*H*-pyrimido[4,5-b]indole (222 mg, 1.0 mmol) and 1-amino-5-methyl-pyrrolidin-2-one (196 mg, 1.3 mmol) in a mixture of dioxane*HCl/IPA (1.5 ml/1.5 ml) was stirred at 95°C overnight. Upon completion, the mixture was cooled to rt and concentrated under reduced pressure. The crude material was purified by HPLC (40–90% $H_2O$/MeOH/0.005% HCl) to afford 1-((7-fluoro-9*H*-pyrimido[4,5-b]indol-4-yl)amino)-5-methylpyrrolidin-2-one (HCl salt, 0.155 g, 46% yield). It was further subjected to chiral chromatography (hexane (0.3% DEA): IPA:MeOH, 90:5:5) to obtain **AVI-3765** (39 mg, retention time = 46.18 min, 99% optic ee) and **AVI-3764** (36 mg, retention time = 51.98 min, 90% optic ee).

### AVI-3765
$^1$H NMR (500 MHz, DMSO) δ 12.22 (s, 1H), 9.31 (s, 1H), 8.47–8.3 (m, 2H), 7.25 (dd, *J* = 9.6, 1.9 Hz, 1H), 7.18–7.11 (m, 1H), 4.03 (s, 1H), 2.39–2.26 (m, 3H), 1.7–1.63 (m, 1H), 1.21 (d, *J* = 6 Hz, 3H). LC–MS (ESI): *m/z* = 300 (M+H)$^+$.

### AVI-3764
$^1$H NMR (500 MHz, DMSO) δ 12.22 (s, 1H), 9.31 (s, 1H), 8.45–8.37 (m, 1H), 8.35 (s, 1H), 7.25 (dd, *J* = 9.4, 2.3 Hz, 1H), 7.2–7.04 (m, 1H), 4.11–3.93 (m, 1H), 2.42–2.19 (m, 3H), 1.75–1.57 (m, 1H), 1.21 (d, *J* = 6.3 Hz, 3H). LC–MS (ESI): *m/z* = 300 (M+H)$^+$.

## AVI-4636

**Chemical structure 9.** AVI-4636.

A mixture of 4-chloro-7-fluoro-9*H*-pyrimido[4,5-b]indole (25 mg, 0.11 mmol) and 1-amino-5,5-dimethylpyrrolidin-2-one hydrochloride salt (28 mg, 0.17 mmol) in isopropanol/aqueous 1 N HCl (2: 1, 0.6 ml) was heated to 100°C for 18 hr. The reaction mixture was purified by reverse phase chromatography (water/acetonitrile/0.1% formic acid) to obtain 10 mg (25%) of 1-((7-fluoro-9*H*-pyrimido[4,5-b]indol-4-yl)amino)-5,5-dimethylpyrrolidin-2-one formic acid salt (**AVI-4636**) as a white solid. $^1$H NMR (METHANOL-d$_4$, 400 MHz) δ 8.30 (s, 1H), 8.09 (dd, 1H, *J* = 5.1, 8.8 Hz), 7.21 (dd, 1H, *J* = 2.3, 9.4 Hz), 6.97 (t, 1H, *J* = 9.2 Hz), 2.59–2.63 (m, 2H), 2.20 (br s, 2H), 1.38 (s, 6H). LC–MS (ESI): *m/z* = 314 (M+H)$^+$.

AVI-4206

**Chemical structure 10.** AVI-4206.

A solution of 3-fluoro-2-nitroaniline (25.00 g, 160 mmol) in THF (500 ml) was added triethylamine (48 g, 480 mmol) and triphosgene (14.2 g, 48 mmol) at 0°C. After stirring for an hour, ethylamine as 2.0 M solution in THF (200 ml) was added. Upon completion of the reaction, the mixture was poured into 500 ml of water, extracted with ethyl acetate (3 × 500 ml), the combined organic layers were washed with brine (500 ml), dried over sodium sulfate, filtered, concentrated under reduced pressure, and the residue was purified by silica gel column chromatography (0–20% ethyl acetate/hexanes) to afford 1-ethyl-3-(3-fluoro-2-nitrophenyl)urea as yellow solid (22.00 g, yield: 60.57%). LC–MS (ESI): $m/z$ = 228.1 (M+H)$^+$.

To a solution of 1-ethyl-3-(3-fluoro-2-nitrophenyl)urea (48 g, 211.45 mmol) in DMF (300 ml) were added methyl 2-isocyanoacetate (41.86 g, 422.90 mmol) and potassium carbonate (87.54 g, 634.36 mmol). The solution was stirred at 80°C for 16 hr. The mixture was adjusted to be weakly acidic by 2 N HCl, extracted with ethyl acetate (500 ml *3), the combined organic layers were washed with brine (300 ml), dried over sodium sulfate, filtered, and concentrated under reduced pressure, the residue was purified via column chromatography on silica gel (0–20% ethyl acetate/hexanes) to afford methyl 2-cyano-2-(3-(3-ethylureido)-2-nitrophenyl)acetate as yellow solid (44.3 g, yield: 68.46%). LC–MS (ESI): $m/z$ = 307.2 (M+H)$^+$.

A mixture of methyl 2-cyano-2-(3-(3-ethylureido)-2-nitrophenyl)acetate (42 g, 137.25 mmol) and acetic acid (250 ml) was heated to 40°C. Zinc (89.75 g, 1372.54 mmol) was then added in portions at a rate such that the reaction temperature did not rise above 60°C. After the addition was complete, the reaction mixture was stirred at 60°C for 2 hr. The reaction mixture was cooled to room temperature and filtered through a celite pad. The filtrate was concentrated under vacuum. The crude product was purified via column chromatography on silica gel (DCM: MeOH = 10:1) to give methyl 2-amino-7-(3-ethylureido)-1H-indole-3-carboxylate as a white solid (16 g, yield: 42.23%). LC–MS (ESI): $m/z$ = 277.2 (M+H)$^+$.

Methyl 2-amino-7-(3-ethylureido)-1H-indole-3-carboxylate (2.0 g, 7.25 mmol) and formamidine acetate (4.53 g, 43.48 mmol) were heated to 140°C for 1 hr. The mixture was cooled to room temperature and diluted with approximately 100 ml of water. The resulting mixture was stirred for 15 min before the solid was collected by filtration. The residue was triturated with DMSO and filtered to afford 1-ethyl-3-(4-hydroxy-9H-pyrimido[4,5-b]indol-8-yl)urea as an off-white solid (1.0 g, yield: 50.8%). $^1$H NMR (500 MHz, DMSO) δ 12.21 (s, 1H), 11.76 (s, 1H), 8.34 (s, 1H), 8.13 (d, $J$ = 3.5 Hz, 1H), 7.64 (d, $J$ = 7.7 Hz, 1H), 7.40 (d, $J$ = 7.3 Hz, 1H), 7.13 (t, $J$ = 7.8 Hz, 1H), 6.28 (t, $J$ = 5.5 Hz, 1H), 3.26–3.11 (m, 2H), 1.10 (t, $J$ = 7.2 Hz, 3H). LC–MS (ESI): $m/z$ = 272.3 (M+H)$^+$.

To a solution of 1-ethyl-3-(4-hydroxy-9H-pyrimido[4,5-b]indol-8-yl)urea (500 mg, 1.85 mmol) in THF (20 ml) was added di-tert-butyl dicarbonate (1.21 g, 5.54 mmol), DIPEA (955 mg, 7.4 mmol), and DMAP (226 mg, 1.85 mmol). The mixture was stirred at room temperature for 16 hr. The mixture was then concentrated under reduced pressure to give crude tert-butyl 4-((tert-butoxycarbonyl)oxy)-8-(3-ethylureido)-9H-pyrimido[4,5-b]indole-9-carboxylate as a yellow oil. It was used in the next step without any purification.

A solution of tert-butyl 4-((tert-butoxycarbonyl)oxy)-8-(3-ethylureido)-9H-pyrimido[4,5-b]indole-9-carboxylate (crude) in POCl₃ (10 ml) was stirred at 90°C for 30 min. The solution was concentrated

under reduced pressure and diluted with acetonitrile, then the pH was adjusted to 7.0 with ammonium hydroxide slowly. The resulting solid was filtered with a vacuum filter and washed with water to give the 1-(4-chloro-9H-pyrimido[4,5-b]indol-8-yl)-3-ethylurea (230 mg, two steps yield: 43.1%) as a light yellow solid. LC–MS (ESI): $m/z$ = 290.2 (M+H)$^+$.

To a solution of 1-(4-chloro-9H-pyrimido[4,5-b]indol-8-yl)-3-ethylurea (290 mg, 1.0 mmol) in dry DMSO (6.0 ml) was added 1-amino-5,5-dimethylpyrrolidin-2-one (192 mg, 1.5 mmol), Pd$_2$(dba)$_3$ (92 mg, 0.1 mmol), tri-tert-butylphosphine tetrafluoroborate (44 mg, 0.15 mmol) and t-BuONa (240 mg, 2.5 mmol). After stirring at 100°C for 8 hr, the reaction mixture was filtered and the filtrate was purified by reversed phase chromatography (water/acetonitrile/0.1% TFA). After lyophilization, then silica gel column chromatography (DCM/MeOH = 10/1) to obtain 1-(4-((2,2-dimethyl-5-oxopyrrolidin-1-yl)amino)-9H-pyrimido[4,5-b]indol-8-yl)-3-ethylurea (**AVI-4206**) (120 mg, yield: 31.5%) as a white solid. $^1$H NMR (400 MHz, DMSO) δ 11.67 (s, 1H), 9.06 (s, 1H), 8.37 (d, $J$ = 15.7 Hz, 2H), 8.07 (d, $J$ = 7.6 Hz, 1H), 7.59 (d, $J$ = 7.8 Hz, 1H), 7.20 (t, $J$ = 7.9 Hz, 1H), 6.28 (t, $J$ = 5.4 Hz, 1H), 3.27–3.10 (m, 2H), 2.42 (t, $J$ = 7.8 Hz, 2H), 2.03 (t, $J$ = 7.8 Hz, 2H), 1.26 (d, $J$ = 21.1 Hz, 6H), 1.11 (t, $J$ = 7.2 Hz, 3H). $^{13}$C NMR (DMSO-d$_6$, 100 MHz) δ 171.8, 157.8, 155.9, 155.8, 154.9, 128.6, 125.4, 125.3, 121.2, 120.4, 117.2, 117.1, 96.6, 61.1, 34.8, 34.7, 32.4, 27.8, 26.6, 15.9. LC–MS (ESI): $m/z$ = 382 (M+H)$^+$.

## In vitro validation
### X-ray crystallography

Mac1 crystals (P4$_3$ construct, residues 3–169) were grown by sitting-drop vapor diffusion in 28% wt/vol polyethylene glycol (PEG) 3000 and 100 mM N-cyclohexyl-2-aminoethanesulfonic acid (CHES) pH 9.5 as described previously (*Schuller et al., 2021*; *Gahbauer et al., 2023*). Compounds prepared in DMSO (100 mM) were added to crystal drops using an Echo 650 acoustic dispenser (*Collins et al., 2017*) (final concentration of 10 mM). Crystals were incubated at room temperature for 2–4 hr prior to vitrification in liquid nitrogen without additional cryoprotection. X-ray diffraction data were collected at the Advanced Light Source (ALS beamline 8.3.1) or the Stanford Synchrotron Light Source (SSRL beamline 9–2). Data were indexed, integrated, and scaled with XDS (*Kabsch, 2010*) and merged with Aimless (*Evans and Murshudov, 2013*). The P4$_3$ Mac1 crystals contain two copies of the protein in the asymmetric unit (chains A and B). The active site of chain A is open; however, chain B is blocked by a crystal contact. We previously observed that potent Mac1 inhibitors dissolve crystals, likely through the displacement of the B chain crystal contact (*Gahbauer et al., 2023*). In addition, crystal packing in the chain A active site restricts movement of the Ala129-Gly134 loop, leading to decreased occupancy for compounds with substituents on the pyrrolidinone. To aid modeling the resulting conformational and compositional disorder, we used the PanDDA method (*Pearce et al., 2017*) to model ligands where the occupancy was low (<25%, AVI-4051, AVI-3367, AVI-3763, AVI-3762, AVI-3765, and AVI-3764) or where there was substantial disorder (AVI-4636). After modeling ligands, structures were refined using phenix.refine (*Liebschner et al., 2019*) as described previously (*Gahbauer et al., 2023*). Data collection settings and statistics are reported in *Supplementary file 1*.

To achieve higher ligand occupancy for AVI-4206, we co-crystallized an alternative Mac1 construct previously reported to crystallize in P1, P2$_1$, and C2 (residues 2–170) (*Michalska et al., 2020*; *Correy et al., 2022*). Crystals grew by sitting-drop vapor diffusion in 200 mM lithium acetate and 20% wt/vol PEG 3350 with 30 mg/ml Mac1 (1.6 mM) and 3.2 mM AVI-4206 (3.2% DMSO). Crystals were vitrified directly in liquid nitrogen, and diffraction data to 0.8 Å were collected at the ALS (beamline 8.3.1). Data were reduced in P1 using the same pipeline as the P4$_3$ crystals. Solvent content analysis suggested that there were two chains in the asymmetric unit. Phases were obtained using Phaser (*McCoy et al., 2007*) and apo Mac1 coordinates (PDB code 7KQO, chain A). Structural refinement was performed with phenix.refine following the previously described procedures for ultra-high-resolution data (*Correy et al., 2022*). After several rounds of refinement, positive difference density was clear for a second, relatively low occupancy, conformation of the entire chain A and B, each representing a ~3.1 Å translation relative to the major conformation. Modeling and inspection of the minor conformations suggested that they cannot be occupied simultaneously; therefore, they were modeled with distinct alternative location identifiers (altlocs). The major conformation (protein, AVI-4206, and water) was modeled with altloc A and the minor conformations (protein, AVI-4206, and water) were modeled with altlocs C and D. In addition to the rigid body disorder, there was clear density for a third conformation of the residue 57–75 α-helix. In chain A, this was modeled with altloc B, while the density in

chain B was too weak to allow modeling. The $F_O - F_C$ difference electron density maps or PanDDA event maps used to model ligands are shown in *Figure 1—figure supplement 1*.

## Inhibition assay

Inhibition of Mac1 ADP-ribosylhydrolase activity by AVI-219 and AVI-4206 was determined using the NUDT5/AMP-Glo assay (*Dasovich et al., 2022*; *Taha et al., 2023b*; *Voorneveld et al., 2018*). The substrate for the reaction was human PARP10 (catalytic domain, residues 819–1007), purified and auto-mono-ADP-ribosylated using NAD+ as described previously (*Taha et al., 2023b*). Briefly, AVI-219 and AVI-4206 were dispensed into 384-well white assay plates (Corning, 3824) using an Echo 650 acoustic dispenser to achieve a final concentration range from 1 mM to 0.4 nM (8 µl reaction volume, 1% DMSO). Purified Mac1 (P4$_3$ construct, 2 µl, 10 nM final concentration) and NUDT5 (2 µl, 100 nM final concentration) were added to wells and the plates were incubated for 5 min at room temperature. Mono-ADP-ribosylated PARP10 was added to wells (4 µl, 2 µM final concentration) and the plates were incubated at room temperature for an additional hour. The concentration of AMP was measured with an AMP-Glo assay kit (Promega, V5011) following the manufacturer's instructions using a BioTek Synergy HTX plate reader. Percentage inhibition was calculated relative to control wells containing no inhibitor (DMSO only, 0% inhibition) or no Mac1 (100% inhibition) and IC$_{50}$ values were determined by fitting a four-parameter sigmoidal dose–response equation using GraphPad Prism (version 10.1.1), with the top and bottom constrained to 100 and 0% inhibition, respectively. Data are presented as the mean ± SD of four technical replicates. A control reaction with increasing concentrations of Mac1 indicated that <50% of the mono-ADP-ribosylated PARP10 was hydrolyzed in the 0% inhibition control (*Figure 1—figure supplement 2*). In addition, a counterscreen to test for NUDT5 inhibition or assay interference was performed with identical reactions, except Mac1 was omitted and ADP-ribose was added to a final concentration of 2 µM (Sigma, A0752).

Inhibition of PARP14 macrodomains was tested with the same protocol as Mac1 with minor adjustments as follows. PARP14 macrodomain activity was tested from 8 to 800 nM. Purified PARP14 MD1-MD2 (residues 784–1196) and PARP14 MD2 (residues 999–1196, serving as a negative control because only MD1 is catalytically active) were added to each well with NUDT5 (2.5 µl, NUDT5: 100 nM final concentration) and the plates were incubated for 5 min at room temperature. Mono-ADP-ribosylated PARP10 was dispensed using Dragonfly (1.5 µM, 2.5 µl) and the plates were incubated at room temperature for an additional hour. The concentration of AMP was measured with an AMP-Glo assay kit (Promega, V5011). Inhibition by AVI-4206 was tested at 50 nM enzyme concentration for PARP14 MD1-2, PARP14 MD2, MAC1 WT, and MAC1 N40D. Enzymes were added to the plate at 50 nM final concentration with NUTD5 at 100 nM final concentration (2 µl). AVI-4206 was dispensed using Integra electronic repeat dispense pipette from 1 mM to 0.4 nM (8 µl reaction volume, 1% DMSO). Mono-ADP-ribosylated PARP10 was dispensed as previously (1.5 µM, 4 µl per well) incubated for an additional hour at room temperature followed by detection of AMP concentration with an AMP-Glo assay kit (Promega, V5011) following the manufacturer's instructions using a BioTek Synergy HTX plate reader.

## HTRF

Binding of the compounds to macrodomain proteins was assessed by the displacement of an ADPr-conjugated biotin peptide from His$_6$-tagged protein using an HTRF-technology-based screening assay which was performed as previously described (*Schuller et al., 2021*). The protein sequences used for SARS-CoV-2 Mac1 and the human macrodomains TARG1 and MacroD2 are listed in *Supplementary file 3*. All proteins were expressed and purified as described previously for SARS-CoV-2 Mac1 (*Schuller et al., 2021*). Compounds were dispensed into ProxiPlate-384 Plus (PerkinElmer) assay plates using an Echo 650 Liquid Handler (Beckman Coulter). Binding assays were conducted in a final volume of 16 µl with 12.5 nM NSP3 Mac1 protein, 200 nM peptide ARTK(Bio)QTARK(Aoa-RADP)S (Cambridge Peptides), 1:20,000 Eu$^{3+}$ cryptate conjugated to a His$_6$-specific antibody (HTRF donor, PerkinElmer AD0402) and 1:500 Streptavidin-XL665 (HTRF acceptor, PerkinElmer 610SAXLB) in assay buffer (25 mM 4-(2-hydroxyethyl)-1-piperazine-1-ethanesulfonic acid (HEPES) pH 7.0, 20 mM NaCl, 0.05% bovine serum albumin and 0.05% Tween-20). TARG1 and MacroD2 binding were measured at 25 nM and 12.5 nM, respectively. Assay reagents were dispensed manually into plates using an electronic multichannel pipette. Macrodomain protein and peptide were dispensed and preincubated

for 30 min at room temperature before HTRF reagents were added. Fluorescence was measured after a 1-hr incubation at room temperature using a PerkinElmer EnVision 2105-0010 Dual Detector Multimode microplate reader with dual emission protocol (A = excitation of 320 nm, emission of 665 nm, and B = excitation of 320 nm, emission of 620 nm). Compounds were tested in triplicate in a 14-point dose response. Raw data were processed to give an HTRF ratio (channel A/B × 10,000), which was used to generate $IC_{50}$ curves. The $IC_{50}$ values were determined by non-linear regression using GraphPad Prism (version 10.1.1). Data are presented as mean ± SD of three technical replicates.

## CETSA

Cellular target engagement of compounds was assessed using a CETSA-nLuc or CETSA-WB assay (*Martinez et al., 2018*). The SARS-CoV-2 Mac1 macrodomain was cloned into pcDNA3.1 by Genscript with both an N-terminal 3XFLAG tag and a C-terminal HiBiT tag as listed in *Supplementary file 3*. A 2A mKate was included to identify successfully transfected cells (e.g., pcDNA-3xFLAG-Mac1$^{WT}$-nLuc-t2A-mKate2). Plasmids were reverse transfected into A549 cells using Lipofectamine 3000 transfection reagent (Thermo). After 48 hr, cells were harvested by trypsinization and resuspended at $1 \times 10^6$ cells/ml in CETSA buffer (1× DPBS (with $CaCl_2$ and $MgCl_2$), 1 g/l glucose and 1× protease inhibitor cocktail Roche, 5892970001). Cells were treated in microcentrifuge tubes with compound or DMSO and incubated at 37°C for 1 hr.

For CETSA-nLuc experiments, 30 µl of suspended cells were dispensed into a 96-well PCR plate (Bio-Rad) and heated for 3.5 min using a preheated gradient thermal cycler (Eppendorf). A Nano-Glo HiBiT Lytic Detection System (Promega) was used to quantify HiBiT-tagged proteins in cell lysates. 30 µl of a mixture containing Lytic Buffer, LgBiT protein, HiBiT Lytic Substrate were added to the cell suspension, and luminescence intensity was measured using a Biotek Synergy H1. Luminescence values for each sample were normalized to the lowest temperature on the range and $T_{agg}$ ($T$-aggregate) values were determined by fitting data with a four-parameter sigmoidal dose–response equation using non-linear regression in GraphPad Prism (version 10.1.1). Delta values were calculated by subtracting the $T_{agg}^{DMSO}$ value from the $T_{agg}^{drug}$ value. Data are presented as mean ± SD of two technical replicates.

For CETSA-WB experiments, 45 µl of suspended cells were dispensed into PCR strip tubes and heated for 3.5 min using a pre-heated gradient thermal cycler (Eppendorf). Samples were then placed in an aluminum PCR block on a dry ice/ethanol bath for 3 min, followed by incubating at 37°C for 3 min, and vortexing for 3 s. This freeze–thaw cycle was repeated three more times. Insoluble proteins were transferred to a 1.5-ml microcentrifuge tube, separated by centrifugation (20,000 × $g$, 15 min, 4°C), and 40 µl of supernatant corresponding to soluble proteins was kept for WB. Samples were separated on an SDS–polyacrylamide gel and transferred to a PVDF membrane. The following antibodies were used for immunoblotting: anti-FLAG antibody (Sigma, F1804, 1:1000 overnight), anti-mouse HRP antibody (CST, 7076S, 1:3000 for 1 hr). Images were captured using the Azure c600 Western Blot Imaging System, quantified using ImageJ, and plotted as above.

## Cellular and organoid studies

### SARS-CoV-2 culture

As described in our previous report (*Taha et al., 2023b*), the pBAC SARS-CoV-2 WT (WA1) and N40D mutant constructs on WA1 background were made by co-transfecting them with an N expression vector into BHK-21 cells. Following 3 days of transfection, the cell supernatants were used to infect Vero cells stably expressing TMPRSS2, followed by passaging to achieve a high viral titer. All viruses generated or used in this study were verified by NGS using the ARTIC Network's protocol. A previously reported mNeon SARS-CoV-2 infectious clone (ic-SARS-CoV-2-mNG) (*Xie et al., 2020*) was passaged on Vero-TMPRSS2 and used for Incucyte-based antiviral assays.

### Cells

BHK-21 obtained from ATCC were grown in DMEM (Corning) with 10% fetal bovine serum (FBS) (GeminiBio), 1× Glutamax (Corning), and 1× penicillin–streptomycin (Corning) at 37°C in a 5% $CO_2$ atmosphere. A549-ACE2h cells were generated by stably expressing hACE2 (*Khalid et al., 2024*) and further selecting for high ACE2 expression levels via FACS with Alexa Fluor 647 conjugated to a hACE2-specific antibody (FAB9332R, R&D Systems). These cells were cultured in DMEM supplemented

with 10% FBS, 10 µg/ml blasticidin (Sigma), 1× NEAA (Gibco), and 1% L-glutamine (Corning) at 37°C in a 5% $CO_2$ atmosphere. Vero cells that stably overexpress human TMPRSS2 (Vero TMPRSS2), a gift from the Whelan lab (*Case et al., 2020*), were cultured under the same conditions. Additionally, Vero cells that stably express human ACE2 and TMPRSS2 (VAT), provided by A. Creanga and B. Graham from the NIH, were maintained in DMEM with 10% FBS, 1× penicillin–streptomycin, and 10 µg/ml puromycin at 37°C in a 5% $CO_2$ atmosphere. A549 cells obtained from ATCC were grown in DMEM Glutamax (Gibco) with 10% FBS (GeminiBio), and 1× penicillin–streptomycin (Corning) at 37°C in a 5% $CO_2$ atmosphere.

## ADP-ribose Western Blots
Calu3 cells were obtained from ATCC and cultured in Advanced DMEM (Gibco) supplemented with 2.5% FBS, 1x GlutaMax, and 1x penicillin–streptomycin at 37°C and 5% $CO_2$. $5 \times 10^6$ cells were plated in 15 cm dishes and media was changed every 2–3 days until the cells were 80% confluent. The cells were treated with INFy 50 ng/ml (R&D Systems) w/without AVI-4206 100 µM. After 6 hr, the cells were infected with WA1 or WA1 NSP3 Mac1 N40D at an MOI of 1 for 36 hr. The cells were washed with phosphate-buffered saline (PBS) × 3 and scraped in Pierce IP Lysis Buffer (Thermo Fisher) containing 1x HALT protease and phosphatase inhibitor mix (Thermo Fisher) on ice. The lysate was stored at –80°C until further processing.

The cell lysate was incubated for 5 min at room temperature with recombinant benzonase. Following incubation, the lysate was centrifuged at 13,000 rpm at 4°C for 20 min, and the supernatant was collected. The samples were then boiled for 5 min at 95°C in 1x NuPAGE LDS sample buffer (Invitrogen) with a final concentration of 1X NuPAGE sample reducing agent (Invitrogen). For the detection of ADPr levels in whole-cell lysates, the samples were subjected to SDS–PAGE and immunoblotting. All primary and secondary antibodies pan-ADP-ribose antibody (MABE1016, Millipore), Mono-ADP-ribose antibody (AbD33204, Bio-Rad), HRP-conjugated (Cell signaling), used at a 1:1000 dilution were diluted in 5% non-fat dry milk in TBST. Signals were detected by chemiluminescence (Thermo) and visualized using the ChemiDoc XRS+ System (Bio-Rad). Densitometric analysis was performed using Image Lab (Bio-Rad). Quantification was normalized to Actin. The data are expressed as mean ± SD. Statistical differences were determined using an unpaired *t*-test in GraphPad Prism 10.3.1.

## Human airway organoids
Human lung tissues were used to generate self-organizing 3D HAOs consisting of basal cells, multi-ciliated epithelial cells, mucus-producing secretory cells, and club cells. As described previously (*Suryawanshi et al., 2022*; *Taha et al., 2023b*), the human lung tissues obtained from Matthay lab were dissociated to single cells using enzymatic digestion. The isolated single cells were resuspended in Basement Membrane Extract (BME, R&D Biosystems), to form a BME droplet containing cells which was submerged in HAO medium consisting 1 mM HEPES (Corning), 1× GlutaMAX (Gibco), 1× penicillin–streptomycin (Corning), 10% R-spondin1 conditioned medium, 1% B27 (Gibco), 25 ng/ml noggin (Peprotech), 1.25 mM *N*-acetylcysteine (Sigma-Aldrich), 10 mM nicotinamide (Sigma-Aldrich), 5 nM heregulin-β1 (Peprotech), and 100 µg/ml Primocin (InvivoGen) in DMEM. This HAO medium was also supplemented with 5 µM Y-27632, 500 nM A83-01, 500 nM SB202190, 25 ng/ml FGF7, and 100 ng/ml FGF10 (all obtained from Stem Cell Technologies). After sufficient growth of HAO cells, in order to differentiate the HAO cells, the HAO medium was replaced with equal proportion of HAO medium and PneumaCult-ALI medium (Stem Cell Technologies).

## SARS-CoV-2 replicon assay
The SARS-CoV-2 replicon assay was conducted as described previously (*Taha et al., 2023b*). Briefly, the pBAC SARS-CoV-2 ΔSpike WT or nsp3 Mac1 N40D modified plasmids (40 µg) were transfected into BHK-21 fibroblast cells along with N and S expression vectors (20 µg each) in a 15 $cm^2$ tissue culture dish. The culture media was replaced with fresh growth medium 12 hr post-transfection. The media containing single-round infectious particles was collected and 0.45-µm-filtered 72 hr post-transfection and stored at –80°C until use.

Vero-ACE2-TMPRSS2 (VAT) and A549 ACE2$^h$ cells were plated $2.5 \times 10^4$ cells per well in 96-well plate in media containing 0, 1000, or 10,000 IU/ml of IFN-γ. After 16 hr, the media was replaced with 50 µl media containing 5x the final desired concentration of IFN-γ and AVI-4206. After 2 hr, 200 µl of

supernatant containing WA1 or WA1 nsp3 Mac1 N40D single-round infectious particles was added. After 8 hr, the cells were washed with 200 μl culture medium and 100 μl of culture medium was added. After 16 hr, 50 μl from each well was transferred to a white 96-well plate to measure nanoluciferase activity by adding 50 μl of Nano-Glo luciferase assay buffer and substrate and analyzed on an Infinite M Plex plate reader (Tecan).

## SARS-CoV-2 in vitro antiviral assay

Antiviral activity of compounds was assessed using the Incucyte live-cell analysis system. $2 \times 10^4$ A549-ACE2h cells per well were seeded in Edge 2.0 96-well plates filled with 1.5 ml PBS in the outer moats and 100 μl in-between wells and incubated at 37°C and 5% $CO_2$. The next day, cells were pretreated with compounds for 2 hr, followed by the removal of the compounds and infection with 50 μl of icSARS-CoV-2-mNG at an MOI of 0.1 for 2 hr. Subsequently, virus inoculum was removed and fresh compounds diluted in DMEM (10% FBS, 1% L-glutamine, 1× P/S, 1× NEAA, Incucyte Cytotox Dye) were added. Infected cells were placed in an Incucyte S3 (Sartorius) and infection and cell death were measured over 48 hr at 1-hr intervals using a 10x objective, capturing 3 images per well at each time point under cell maintenance conditions (37°C, 5% $CO_2$). Infection and cell death were quantified as Total Green Object Integrated Intensity (300 ms acquisition time) and Red Object Integrated Intensity (400 ms acquisition time), respectively. After in-built software analysis, raw data was exported and antiviral efficacy was determined as the percentage of viral replication normalized to the vehicle control. Nirmatrelvir (HY-138687, MedChemExpress) and uninfected cells were used as intra-assay positive and negative controls, respectively. Unless otherwise stated, experiments were conducted in triplicate with three technical replicates. $EC_{50}$ values were calculated using GraphPad PRISM 10 (La Jolla, CA, USA) employing a dose–response inhibition equation with a non-linear fit regression model.

## Antiviral efficacy in HAOs

The differentiated HAOs were utilized to analyze the dose-dependent anti-SARS-CoV-2 efficacy of AVI-4206. Briefly, 100,000 cells of differentiated HAOs were seeded in a V-bottom plate (Greiner Bio-One). The cells were pretreated for 2 hr prior to infection with various concentrations of AVI-4206 (0, 0.16, 0.8, 4, 20, and 100 μM). After pretreatment, the HAOs were washed and infected with SARS-CoV-2 WA1 at an MOI of 1. A WA1-N40D mutant strain lacking the macrodomain activity was used as a positive control. Following 2 hr of infection, the HAOs were washed three times. Each washing step involved replacing the media with PBS and centrifuging the cells at 1000 rpm for 3 min. After three washes, the PBS was replaced with 100 μl of HAO differentiation medium, with or without varying concentrations of AVI-4206, and the plate was incubated for 72 hr at 37°C with 5% $CO_2$. Supernatants collected at 24-hr intervals were used to analyze mature virus particle formation via plaque assay.

## Drug cytotoxicity assay

A549-ACE2h cells were seeded and incubated as for the in vitro antiviral assay. Cells were treated with compounds at the respective concentrations and vehicle control for 50 hr at 37°C and 5% $CO_2$. Subsequently, Cell Titer-Glo reagent was added in a 1:1 ratio to the cells and incubated at room temperature for 5 min before transferring 100 μl of the mixture to a white 96-well plate. Luciferase activity was measured using an Infinite M Plex plate reader (Tecan). Cell viability was determined as the percentage of viability normalized to the vehicle control. Compound cytotoxicity was assessed in parallel with infection experiments using cells of the same passage.

## Macrophage infection assay

MDMs were generated from peripheral blood mononuclear cells obtained from healthy donors (Vitalant, CA, USA). CD14+ monocytes were isolated by negative selection using the EasySep Human Monocyte Isolation Kit (STEMCELL Technologies, Vancouver, Canada), and cultured in ImmunoCult-SF Macrophage Medium (STEMCELL Technologies) supplemented with 50 ng/ml M-CSF for 8 days, following the manufacturer's protocol. Differentiated MDMs were pretreated with AVI-4206 at the indicated concentrations in the presence of 50 ng/ml IFN-γ for 2 hr. Cells were then infected with SARS-CoV-2 (strain USA-WA1/2020) at an MOI of 2 for 2 hr. Following infection, cells were washed three times with PBS to remove unbound virus and cultured for an additional 18 hr in medium containing

the corresponding concentrations of AVI-4206 and 50 ng/ml IFN-γ. Viral particle production in the supernatant was quantified by plaque assay. Data were analyzed using GraphPad Prism version 10.2.0 (GraphPad Software, CA, USA).

## A549 Mac1 expression cell construction

Mac1 wild-type (Mac1) and N40D mutant (Mac1 N40D) gene fragments were loaded into pLVX-EF1α-IRES-Puro (empty vector, EV) using Gibson cloning kit (NEB E5510). Lentivirus was prepared as previously described (*Shifrut et al., 2018*). Briefly, 15 million HEK293T cells were grown overnight on 15 cm poly-L-lysine coated dishes and then transfected with 6 µg pMD2.G (Addgene plasmid # 12259; http://n2t.net/addgene:12259; RRID:Addgene_12259), 18 µg dR8.91 (since replaced by second generation compatible pCMV-dR8.2, Addgene plasmid #8455) and 24 µg pLVX-EF1α-IRES-Puro (EV, Mac1, Mac1-N40D) plasmids using the lipofectamine 3000 transfection reagent per the manufacturer's protocol (Thermo Fisher Scientific, Cat #L3000001). pMD2.G and dR8.91 were a gift from Didier Trono. The following day, media was refreshed with the addition of viral boost reagent at 500x as per the manufacturer's protocol (Alstem, Cat #VB100). Viral supernatant was collected 48 hr post-transfection and spun down at 300 × *g* for 10 min to remove cell debris. To concentrate the lentiviral particles, Alstem precipitation solution (Alstem, Cat #VC100) was added, mixed, and refrigerated at 4°C overnight. The virus was then concentrated by centrifugation at 1500 × *g* for 30 min, at 4°C. Finally, each lentiviral pellet was resuspended at 100x of original volume in cold DMEM + 10% FBS + 1% penicillin-streptomycin and stored until use at –80°C. To generate Mac1 overexpressing cells, 2 million A549 cells were seeded in 10 cm dishes and transduced with lentivirus in the presence of 8 µg/ml polybrene (Sigma, TR-1003-G). The media was changed after 24 hr and, after 48 hr, media containing 2 µg/ml puromycin was added. Cells were selected for 72 hr and then expanded without selection. The expression of Mac1 was confirmed by Western Blot.

## Immunofluorescence assay

To assess the effect of Mac1 on IFN-induced ADP-ribosylation. A549-pLVX-EV, A549-pLVX-Mac1, and A549-pLVX-Mac1-N40D cells were seeded in 96-well plate (10,000 cells/well). Cells were pretreated with medium or 100 unit/ml IFN-γ (Sigma, SRP3058) for 24 hr to induce the expression of ADP-ribosylation. These three cell lines were then treated the next day with the indicated concentrations of AVI-4206 or RBN012759 (Medchemexpress, HY-136979). After 24 hr of exposure to drugs, treated cells were fixed in pre-cooled methanol at –20°C for 20 min, blocked in 3% bovine serum albumin for 15 min, incubated with Poly/Mono-ADP Ribose (E6F6A) Rabbit mAb (CST, 83732S) or Poly/Mono-ADP Ribose (D9P7Z) Rabbit mAb (CST, 89190S) antibodies for 1 hr, and then incubated with Goat anti-Rabbit IgG Secondary Antibody, Alexa Fluor 488 (Thermo Fisher, A-11008) secondary antibodies for 30 min and stained with DAPI for 10 min. Fluorescent cells were imaged with an IN Cell Analyzer 6500 System (Cytiva) and analyzed using IN Carta software (Cytiva).

## TPP assay

Pelleted A549 cells were resuspended in extraction buffer 1× PBS + phosphatase and protease inhibitors (phosSTOP (Roche) and cOmplete Mini Protease Inhibitor Cocktail (Roche)) with gentle pipetting followed by rotation at 4°C for 30 min. Lysates were centrifuged at 1000 × *g* for 10 min at 4°C and supernatant was transferred to new tubes. Recombinant Mac1 was spiked into lysate to a final concentration of 0.05 µM. Lysates + Mac1 were incubated with compound at a final concentration of 100 µM AVI-4206 or DMSO for 30 min at 25°C. Lysates (2 replicates per condition) were distributed into 10 aliquots (20 µl each) in PCR tubes. Samples were heated from 37 to 64°C in 3°C increments on a Bio-Rad C1000 Touch Thermal cycler and held for 4 min at the specified temperature. Samples were held at room temperature for 3 min. Samples were then subjected to two cycles of flash freezing and thawing at 35°C. Aggregated proteins were removed by centrifugation at 20,000 g for 60 min. 20 µl of lysis buffer (8 M urea, 100 mM Tris, pH ~7.5) was added to each well and samples were incubated for 30 min at room temperature. Samples were reduced and alkylated by the addition of TCEP (100 mM final) and 2-chloroacetamide (44 mM final) followed by incubation at room temperature for 30 min. Urea concentration was diluted to 1 M by the addition of 100 mM tris (pH ~7.5). Samples were digested overnight with LysC (Wako, 1:100 enzyme:protein ratio) and trypsin (Promega, 1:50 enzyme:protein ratio). Samples were desalted with a 96-well mini 20 MG PROTO 300 C18 plate (HNS

S18V, The Nest Group) according to manufacturer's directions. Peptide concentration was determined by NanoDrop (Thermo).

Following digestion, peptides were injected onto a timsTOF SCP (Bruker) connected to either an EASY-nLC 1200 system (Thermo) or VanquishNeo (Thermo). Peptides were separated on a PepSep reverse-phase C18 column (1.9 µm particles, 15 cm, 150 mm ID) (Bruker) with a gradient of 5–28% buffer B (0.1% formic acid in acetonitrile) over buffer A (0.1% formic acid in water) over 20 min, an increase to 32% B in 3 min, and held at 95% B for 7 min. DIA-PASEF analyses were acquired from 100 to 1700 $m/z$ over a 1/Kø of 0.70–1.30 Vs/cm$^2$, with a ramp and accumulation time set to 75 ms. Library DDA PASEF runs were collected over the same m/z and 1/Kø range and a cycle time of 1.9 s.

All data was searched against the Uniprot Human database (downloaded 05/25/23) appended with the SARS-CoV-2 database (downloaded 02/20/2024) using a combined DDA and DIA library in Spectronaut (Biognosys, version 19.0). Default settings, including trypsin digestion, variable modifications of methionine oxidation and N-termini acetylation, and fixed modification of cysteine carbamidomethylation, were used. Missing values were imputed for each run using background intensity. Data was filtered to obtain a false discovery rate of 1% at the peptide spectrum match and protein level. Lysate experiments were normalized to the lowest temperature (37°C), and melting points were determined in R using the Inflect package (*McCracken et al., 2021*).

## PK and in vivo studies
### ADMET target and kinase studies
The kinase assessment was performed using contract services by Eurofins using their scanEDGE KINOMEscan Assay Platform (Study Code: US073-0032699). Assessment of ADMET targets (cardiac channel profiling, CYP induction, peptidase selectivity panel, and secondary pharmacology profiling) was performed via NIAID's suite of preclinical services for in vitro assessment (Contract No. HHSN27 2201800007I/75N93022F00001).

### PK studies
The PK study of AVI-4206 with IV (10 mg/kg), PO (50 mg/kg), and IP (100 mg/kg) dosing (*Figure 4B* and *Supplementary file 2b*) was performed in male CD1 mice ($n$ = 3 per group) using a formulation of 10% DMSO: 50% PEG 400: 40% of a 20% HP-β-CD in water. Microsampling (40 ml) via facial vein was performed at 0, 0.083, 0.25, 0.5, 1, 2, 4, 8, and 24 hr into K$_2$EDTA tubes. The blood samples were collected and centrifuged to obtain plasma (8000 rpm, 5 min) within 15 min post sampling. Nine blood samples were collected from each mouse; three samples were collected for each time point. Data was processed by Phoenix WinNonlin (version 8.3); samples below the limit of quantitation were excluded in the PK parameters and mean concentration calculation.

## Animal experiments
All the mice experiments were approved (AN169239-01) by the Institutional Animal Care and Use committees at the University of California, San Francisco and Gladstone Institutes and performed in strict accordance with the National Institutes of Health Guide for the Care and Use of Laboratory Animal. For screening of lead macrodomain inhibitors, we employed a transgenic mice model capable of expressing human ACE2. Female mice were divided into three groups: test, positive control, and negative control. The positive control groups were infected (5 × 10$^2$ PFUs) with the N40D mutant of SARS-CoV-2, while the other mice were infected with the WA1 strain. Intraperitoneal treatments were administered twice daily, which began at a day prior to infection and continued until 5 days post-infection, with close monitoring for disease parameters such as weight loss, hypothermia, and hunched posture. At 4 and 7 days post-infection, a subset of mice from each group was euthanized, and their lungs and brain tissues were harvested for virus titration by plaque assay and cytokine expression.

In experiments using wild-type female mice, animals were randomly assigned to one of three groups: an experimental group, a positive control group, and a negative control group. The positive control group was infected intranasally with 1 × 10$^4$ PFU of a mouse-adapted SARS-CoV-2 N40D variant strain, while the experimental and negative control groups were infected with a mouse-adapted SARS-CoV-2 WA1 strain. A mouse-adapted (MA)-SARS-CoV-2 (Spike:Q498Y/P499T) was constructed using pGLUE. Intraperitoneal administration of the test compound began 1 day prior to

infection and continued twice daily for 4 days post-infection. On days 2 and 5 post-infection, subsets of mice from each group were euthanized, and lung tissues were harvested for viral load quantification by plaque assay.

## Plaque assay

The mature virus particles in the lung homogenates were analyzed using plaque assay. Briefly, VAT cells were seeded in a 12-well plate and incubated overnight. The cells were inoculated with $10–10^6$ dilutions of the respective lung homogenates. After 1-hr incubation, the lung homogenates in the wells were overlaid with 2.5% Avicel (RC-591, Dupont). And the plates were incubated at 370°C and 5% $CO_2$ for 48 hr. After incubation, the overlay media was removed and the cells were fixed in 10% formalin. The plaques were visualized by staining the cells with crystal violet. Data analysis was performed by using GraphPad Prism version 10.

## Cytokine estimation

Lung homogenates were clarified by centrifugation at 6000 rpm for 10 min and were used for enzyme-linked immunosorbent assay-based cytokine estimation. The assays were performed as per manufacturer's protocol for IP-10 (Invitrogen, catalog#BMS56018 and BMS6018TEN), IL-2 (Invitrogen, catalog#BMS601, and BMS601TEN), IL-6 (Invitrogen, catalog#BMS103-2, BMS603-2TWO, and BMS603-2TEN), TNF-a (Invitrogen, catalog#BMS607-3 and BMS607-3TEN), and IL1b (Invitrogen, catalog#BMS6002-2 and BMS6002-2TEN).

## Histology

Mouse lung tissues were fixed in 4% PFA (Sigma-Aldrich, Cat #47608) for 24 hr, washed three times with PBS, and stored in 70% ethanol. All the stainings were performed at Histo-Tec Laboratory (Hayward, CA). Samples were processed, embedded in paraffin, and sectioned at 4 μm. The slides were dewaxed using xylene and alcohol-based dewaxing solutions. Epitope retrieval was performed by heat-induced epitope retrieval (HIER) of the formalin-fixed, paraffin-embedded tissue using citrate-based pH 6 solution (Leica Microsystems, AR9961) for 20 min at 95°C. The tissues were stained for H&E, caspase-3 (Biocare #CP229c 1:100), and trichrome, dried, coverslipped (TissueTek-Prisma Coverslipper), and visualized using Axioscan 7 slide scanner (ZEISS) at 40X. Image quantification was performed with ImageJ software and GraphPad Prism.

## Acknowledgements

This work was supported by the National Institutes of Health (NIAID Antiviral Drug Discovery (AViDD)) grant U19AI171110. MO is supported by the James B Pendleton Charitable Trust and the Roddenberry Foundation. The synchrotron X-ray diffraction data used to determine Mac1 structures were collected at beamline 8.3.1 of the Advanced Light Source (ALS) and beamlines 9-2, 12-1, and 12-2 of the Stanford Synchrotron Radiation Lightsource (SSRL). The ALS, a US DOE Office of Science User Facility under Contract No. DE-AC02-05CH11231, is supported in part by the ALS-ENABLE program funded by the NIH, National Institute of General Medical Sciences, grant P30 GM124169. Use of the SSRL, SLAC National Accelerator Laboratory, is supported by the U.S. Department of Energy, Office of Science, Office of Basic Energy Sciences under Contract No. DE-AC02-76SF00515. The SSRL Structural Molecular Biology Program is supported by the DOE Office of Biological and Environmental Research, and by the National Institutes of Health, National Institute of General Medical Sciences (P30GM133894). RKS is supported by the NIH Division of Intramural Research.

This research was supported in part by the intramural research program of the National Institutes of Health (NIH). The contributions of the NIH author were made as a part of their official duties as NIH federal employees, are in compliance with the agency policy requirements, and are considered Works of the United States Government. However, the findings and conclusions presented in this paper are those of the author and do not necessarily reflect the views of the NIH or the U.S. Department of Health and Human Services.

# Additional information

## Competing interests

Rahul K Suryawanshi, Priyadarshini Jaishankar, Galen J Correy, Moira M Rachman, Patrick C O'Leary, Taha Y Taha, Francisco J Zapatero-Belinchón, Morgan E Diolaiti, Mauricio Montano, Takaya Togo, Ryan L Gonciarz: listed as an inventor on a patent application (Mac1 Inhibitors and Uses Thereof U.S. Provisional Application No. 63/631,958 filed April 9, 2024) describing small molecule macrodomain inhibitors, which includes compounds described herein. Nevan J Krogan: The Krogan laboratory has received research support from Vir Biotechnology, F Hoffmann-La Roche and Rezo Therapeutics. NJK has a financially compensated consulting agreement with Maze Therapeutics. He is on the Board of Directors and is President of Rezo Therapeutics and is a shareholder in Tenaya Therapeutics, Maze Therapeutics, Rezo Therapeutics, and GEn1E Lifesciences. He is also listed as an inventor on a patent application (Mac1 Inhibitors and Uses Thereof U.S. Provisional Application No. 63/631,958 filed April 9, 2024) describing small molecule macrodomain inhibitors, which includes compounds described herein. Brian K Shoichet: co-founder of BlueDolphin LLC, Epiodyne Inc, and Deep Apple Therapeutics, Inc, and serves on the SRB of Genentech, the SAB of Schrodinger LLC, and the SAB of Vilya Therapeutics. Also listed as an inventor on a patent application (Mac1 Inhibitors and Uses Thereof U.S. Provisional Application No. 63/631,958 filed April 9, 2024) describing small molecule macrodomain inhibitors, which includes compounds described herein. Melanie Ott: cofounder of Directbio and board member of InVisishield. TYT and MO are listed as inventors on a patent application (Rapid generation of infectious clones US-2024/0209381-A1) filed by the Gladstone Institutes that covers the use of pGLUE to generate SARS-CoV-2 infectious clones and replicons. Also listed as an inventor on a patent application (Mac1 Inhibitors and Uses Thereof U.S. Provisional Application No. 63/631,958 filed April 9, 2024) describing small molecule macrodomain inhibitors, which includes compounds described herein. Adam R Renslo: co-founder of TheRas, Elgia Therapeutics, and Tatara Therapeutics, and receives sponsored research support from Merck, Sharp and Dohme. Listed as an inventor on a patent application (Mac1 Inhibitors and Uses Thereof U.S. Provisional Application No. 63/631,958 filed April 9, 2024) describing small molecule macrodomain inhibitors, which includes compounds described herein. Alan Ashworth: co-founder of Tango Therapeutics, Azkarra Therapeutics and Kytarro; a member of the board of Cytomx, Ovibio Corporation, Cambridge Science Corporation; a member of the scientific advisory board of Genentech, GLAdiator, Circle, Bluestar/Clearnote Health, Earli, Ambagon, Phoenix Molecular Designs, Yingli/280Bio, Trial Library, ORIC and HAP10; a consultant for ProLynx, Next RNA and Novartis; receives research support from SPARC; and holds patents on the use of PARP inhibitors held jointly with AstraZeneca from which he has benefited financially (and may do so in the future); listed as an inventor on a patent application (Mac1 Inhibitors and Uses Thereof U.S. Provisional Application No. 63/631,958 filed April 9, 2024) describing small molecule macrodomain inhibitors, which includes compounds described herein. James S Fraser: consultant to, shareholder of, and receives sponsored research support from Relay Therapeutics, and is listed as an inventor on a patent application (Mac1 Inhibitors and Uses Thereof U.S. Provisional Application No. 63/631,958 filed April 9, 2024) describing small molecule macrodomain inhibitors, which includes compounds described herein. The other authors declare that no competing interests exist.

## Funding

| Funder | Grant reference number | Author |
| --- | --- | --- |
| National Institutes of Health | U19AI171110 | Nevan J Krogan<br>Melanie Ott<br>Adam R Renslo<br>Alan Ashworth<br>James S Fraser |

The funders had no role in study design, data collection, and interpretation, or the decision to submit the work for publication.

## Author contributions

Rahul K Suryawanshi, Priyadarshini Jaishankar, Yusuke Matsui, Investigation, Methodology, Writing – original draft, Writing – review and editing; Galen J Correy, Conceptualization, Formal analysis,

Investigation, Visualization, Methodology, Writing – original draft, Writing – review and editing; Moira M Rachman, Patrick C O'Leary, Taha Y Taha, Conceptualization, Investigation, Methodology, Writing – original draft, Writing – review and editing; Francisco J Zapatero-Belinchón, Maria McCavitt-Malvido, Yagmur U Doruk, Maisie GV Stevens, Morgan E Diolaiti, Manasi P Jogalekar, Huadong Chen, Alicia L Richards, Pornparn Kongpracha, Sofia Bali, Mauricio Montano, Julia Rosecrans, Michael Matthay, Takaya Togo, Ryan L Gonciarz, R Jeffrey Neitz, Investigation, Methodology, Writing – review and editing; Saumya Gopalkrishnan, Conceptualization, Supervision, Writing – review and editing; Nevan J Krogan, Supervision, Funding acquisition, Writing – review and editing; Danielle L Swaney, Formal analysis, Supervision, Investigation, Methodology, Writing – review and editing; Brian K Shoichet, Melanie Ott, James S Fraser, Conceptualization, Supervision, Funding acquisition, Validation, Investigation, Visualization, Methodology, Writing – original draft, Project administration, Writing – review and editing; Adam R Renslo, Alan Ashworth, Conceptualization, Formal analysis, Supervision, Funding acquisition, Validation, Investigation, Visualization, Methodology, Writing – original draft, Project administration, Writing – review and editing

### Author ORCIDs

Rahul K Suryawanshi ⓘ https://orcid.org/0000-0001-8374-669X
Priyadarshini Jaishankar ⓘ https://orcid.org/0009-0005-2013-8941
Patrick C O'Leary ⓘ https://orcid.org/0000-0002-2919-5943
Taha Y Taha ⓘ https://orcid.org/0000-0002-7344-7490
Yusuke Matsui ⓘ https://orcid.org/0000-0002-6016-2867
Francisco J Zapatero-Belinchón ⓘ https://orcid.org/0000-0002-2751-8411
Morgan E Diolaiti ⓘ https://orcid.org/0000-0001-5900-3060
Huadong Chen ⓘ https://orcid.org/0000-0003-4681-0853
Pornparn Kongpracha ⓘ https://orcid.org/0000-0003-1759-213X
Sofia Bali ⓘ http://orcid.org/0000-0002-4046-7081
Mauricio Montano ⓘ https://orcid.org/0000-0002-0353-0037
Michael Matthay ⓘ https://orcid.org/0000-0003-3039-8155
Takaya Togo ⓘ https://orcid.org/0000-0003-0243-0760
Saumya Gopalkrishnan ⓘ https://orcid.org/0009-0003-6713-4492
R Jeffrey Neitz ⓘ https://orcid.org/0000-0002-2247-9345
Nevan J Krogan ⓘ https://orcid.org/0000-0003-4902-337X
Danielle L Swaney ⓘ https://orcid.org/0000-0001-6119-6084
Brian K Shoichet ⓘ https://orcid.org/0000-0002-6098-7367
Melanie Ott ⓘ https://orcid.org/0000-0002-5697-1274
Adam R Renslo ⓘ https://orcid.org/0000-0002-1240-2846
James S Fraser ⓘ https://orcid.org/0000-0002-5080-2859

Reviewer #1 (Public review): https://doi.org/10.7554/eLife.103484.3.sa1
Reviewer #2 (Public review): https://doi.org/10.7554/eLife.103484.3.sa2
Reviewer #3 (Public review): https://doi.org/10.7554/eLife.103484.3.sa3
Author response https://doi.org/10.7554/eLife.103484.3.sa4

## Additional files

### Supplementary files

Supplementary file 1. X-ray data collection and refinement deposition statistics.

Supplementary file 2. Specificity, PK, and ADMET panels. (a) Eurofins scanEDGE kinase assay shows no inhibition greater than >35% at 10 µM across a panel of diverse kinases. (b) Pharmacokinetic parameters for AVI-4206 following IV (10 mg/kg), PO (50 mg/kg), and IP (100 mg/kg) doses in male CD1 mice ($n$ = 3 per group). (c) ADMET panel shows no antagonist response greater than >15% at 10 µM.

Supplementary file 3. Macrodomain protein sequences.

MDAR checklist

## Data availability

X-ray structures have been deposited in the Protein Data Bank as: 9CXY (AVI-1500), 9CXZ (AVI-1501), 7HC4 (AVI-3367), 7HC5 (AVI-3765), 7HC6 (AVI-3764), 7HC7 (AVI-4051), 7HC8 (AVI-3763), 7HC9 (AVI-3762), 7HCA (AVI-4636), 9CY0 (AVI-4206). All other data supporting the findings of the present study are available in the article, extended data, and figure supplements.

The following datasets were generated:

| Author(s) | Year | Dataset title | Dataset URL | Database and Identifier |
|---|---|---|---|---|
| Correy GJ, Fraser JS | 2025 | 9CXY - Crystal structure of SARS-CoV-2 NSP3 macrodomain in complex with AVI-1500 | https://doi.org/10.2210/pdb9cxy/pdb | Worldwide Protein Data Bank, 10.2210/pdb9cxy/pdb |
| Correy GJ, Fraser JS | 2025 | 9CXZ - Crystal structure of SARS-CoV-2 NSP3 macrodomain in complex with AVI-1501 | https://doi.org/10.2210/pdb9cxz/pdb | Worldwide Protein Data Bank, 10.2210/pdb9cxz/pdb |
| Correy GJ, Fraser JS | 2025 | PanDDA analysis group deposition -- Crystal structure of SARS-CoV-2 NSP3 macrodmain in complex with AVI-3367 | https://doi.org/10.2210/pdb7hc4/pdb | Worldwide Protein Data Bank, 10.2210/pdb7hc4/pdb |
| Correy GJ, Fraser JS | 2025 | PanDDA analysis group deposition -- Crystal structure of SARS-CoV-2 NSP3 macrodmain in complex with AVI-3765 | https://doi.org/10.2210/pdb7hc5/pdb | Worldwide Protein Data Bank, 10.2210/pdb7hc5/pdb |
| Correy GJ, Fraser JS | 2025 | PanDDA analysis group deposition -- Crystal structure of SARS-CoV-2 NSP3 macrodmain in complex with AVI-3764 | https://doi.org/10.2210/pdb7hc6/pdb | Worldwide Protein Data Bank, 10.2210/pdb7hc6/pdb |
| Correy GJ, Fraser JS | 2025 | PanDDA analysis group deposition -- Crystal structure of SARS-CoV-2 NSP3 macrodmain in complex with AVI-4051 | https://doi.org/10.2210/pdb7hc7/pdb | Worldwide Protein Data Bank, 10.2210/pdb7hc7/pdb |
| Correy GJ, Fraser JS | 2025 | PanDDA analysis group deposition -- Crystal structure of SARS-CoV-2 NSP3 macrodmain in complex with AVI-3763 | https://doi.org/10.2210/pdb7hc8/pdb | Worldwide Protein Data Bank, 10.2210/pdb7hc8/pdb |
| Correy GJ, Fraser JS | 2025 | PanDDA analysis group deposition -- Crystal structure of SARS-CoV-2 NSP3 macrodmain in complex with AVI-3762 | https://doi.org/10.2210/pdb7hc9/pdb | Worldwide Protein Data Bank, 10.2210/pdb7hc9/pdb |
| Correy GJ, Fraser JS | 2025 | PanDDA analysis group deposition -- Crystal structure of SARS-CoV-2 NSP3 macrodmain in complex with AVI-4636 | https://doi.org/10.2210/pdb7hca/pdb | Worldwide Protein Data Bank, 10.2210/pdb7hca/pdb |
| Correy GJ, Fraser JS | 2025 | PDB Entry - 9CY0 (pdb_00009cy0) | https://doi.org/10.2210/pdb9cy0/pdb | Worldwide Protein Data Bank, 10.2210/pdb9cy0/pdb |

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
