## [Editor Report · eLife Assessment]

This **important** study presents the development of a novel inhibitor for SARS-CoV-2 Mac1 that has potential utility both as an antiviral therapeutic and as a tool for probing the molecular mechanisms by which infection-induced ADP-ribosylation triggers robust host antiviral responses. Though minor gaps in understanding the compound's precise molecular mechanism of action and its ability to target Mac1 from other coronaviruses remain, the evidence for its effects on SARS-CoV-2 in relevant biological models is **compelling**.

---

## [Referee Report · Reviewer #1 (Public review)]

SARS-CoV-2 encodes a macrodomain (Mac1) within the nsp3 protein that removes ADP-ribose groups from proteins. However, its role during infection is not well understood. Evidence suggests that Mac1 antagonizes the host interferon response by counteracting the wave of ADP ribosylation that occurs during infection. Indeed, several PARPs are interferon-stimulated genes. While multiple targets have been proposed, the mechanistic links between ADP ribosylation and a robust antiviral response remain unclear.

Genetic inactivation of Mac1 abrogates viral replication in vivo, suggesting that small-molecule inhibitors of Mac1 could be developed into antivirals to treat COVID-19 and other emerging coronaviruses. The authors report a potent and selective small molecule inhibitor targeting Mac1 (AVI-4206) that demonstrates efficacy in human airway organoids and animal models of SARS-CoV-2 infection. While these results are compelling and provide proof of concept for the therapeutic targeting of Mac1, I am particularly intrigued by the potential of this compound as a probe to elucidate the mechanistic connections between infection-induced ADP ribosylation and the host antiviral response.

The precise function of Mac1 remains unclear. Given its presence in multiple viruses, it likely acts on a fundamental host immune pathway(s). AVI-4206, while promising as a lead compound for the development of antivirals targeting coronaviruses, could also be a valuable tool for uncovering the function of the Mac1 domain. This may lead to fundamental insights into the host immune response to viral infection.

---

## [Referee Report · Reviewer #2 (Public review)]

Summary:

The authors describe the development of a novel inhibitor (AVI-4206) for the first macrodomains of the nsp3 protein of SARS-CoV-2 (Mac1). This involves both medical chemical synthesis, structural work as well as biochemical characterisation. Subsequently the authors present their finding of the efficacy of the inhibitor both on cell culture as well as animal models of SARS-CoV-2 infection. They find that despite high affinity for Mac1 and the known replicatory defects of catalytically inactive Mac1 only moderate beneficial effects can be observed in their chosen models.

Strengths:

The authors employ a variety of different assay to study the affinity, selectivity and potency of the novel inhibitor and thus the in vitro data are very compelling.

Similarly, the authors use several cell culture and in vivo models to strengthen their findings. In addition, the authors address several aspects of the health impact of coronaviral infections from animal survival, over viral load to histological assessment of lung damage.

Weaknesses:

The selection of Targ1 and MacroD2 as off-target human macrodomains is sub-optimal as several studies have shown that the first macrodomains of PARP9 and PARP14 are much closer related to coronaviral macrodomains and both macrodomains are implicated in antiviral defence and immunity. However, the authors address this issue by providing modeling data that show clashes with AVI-4206 similarly to their models with MacroD2 and TARG1.

Comments on revisions:

While the authors have not addressed all my suggestions experimentally, I would like to nevertheless congratulate them on a significantly strengthened manuscript that will provide a valuable contribution to the field.

---

## [Referee Report · Reviewer #3 (Public review)]

Summary:

The authors were trying to validate SARS-CoV-2 Mac1 as a drug discovery target and by extension other viral macrodomains.

Strengths:

The medicinal chemistry and structure based optimization is exemplary. Macrodomains and ADPribosyl hydrolases have the reputation for being undruggable, yet the authors managed to optimize hits from a fragment screen using structure based approaches and fragment linking to make a 20nM inhibitor as a tool compound to validate the target.

In addition, the in vivo work is also a strength. The ability to reduce the viral count at a rate comparable to nirmatrelvir is impressive. Tracking the cytokine expression levels also supports much of the genetic data and mechanism of action for macrodomains.

Weaknesses:

The main compound AVI-4206, while being very potent and selective is not appreciably orally bioavailable. The fact that they have to use high doses of the compound IP to see in vivo effects may lead to questions regarding off target effects. The authors acknowledge this and point it out as a potential avenue for further optimization.

The cellular models are not as predictive of antiviral activity as one would expect. However, the authors had enough chutzpah to test the compound in vivo knowing that cellular models might not be an accurate representation of a living system with a fully functional immune system all of which is most likely needed in an antiviral response to test the importance of Mac1 as a target.

Comments on revisions:

All previous suggestions were addressed. I am satisfied with the author's modifications.

---

## [Author Response]

The following is the authors’ response to the previous reviews.

**Reviewer #1 (Recommendations for the authors):**
Although this study is rigorous and the paper is well-written, I have a few concerns that the authors should address before publication.(1) Cellular levels of protein ADP-ribosylation should be analyzed using anti-ADPR antibodies following infection, both with and without Mac1 and AVI-4206 treatment. While the authors have provided impressive in vivo data, these experiments could ideally be conducted in mice. However, I would be amenable to these analyses being performed in human airway organoids, as they demonstrate clear phenotypes following AVI-4206 treatment post-infection. For a more in-depth exploration, the authors could consider affinity purifying ADP-ribosylated proteins and identifying them via mass spectrometry. I would find it particularly compelling if this approach revealed components of the NF-kB signaling pathway, given the intriguing results presented in Fig. 5. I am also curious if there are differences in ADP ribosylated proteins when comparing Mac1 KO SARS-C0V-2 to AVI-4206 treatment.

We note that despite the recent flurry of activity around Mac1, there is a surprising lack of public data on overall ADPr levels or targets. While we will address the literature precedence for PARP14 signals specifically below (**Reviewer 2 point (h)**) by immunofluorescence, we note that overall levels have not been characterized biochemically previously. Recent PARP14 papers and the ASAP AViDD preprint show changes by immunofluorescence only: and the evidence in that preprint is quite modest - see Figure 7B - https://pmc.ncbi.nlm.nih.gov/articles/PMC11370477/.

We suspect the difficulty in tracking changes biochemically is due to multiple factors that influence the overall detectability and reproducibility. First, with regard to detectability - it is quite possible that only a small change in the ADPr status of a small number of targets is responsible for the phenotypes in vivo. Virus levels are very low in the organoid system and the variability in ADPr levels from tissue samples from in vivo experiments is high. Given the difficulty in translating back to cellular models, this problem is therefore magnified further. Second, with regard to reproducibility - we observe a great deal of reagent dependence on ADPr signals by Western blot+/- Mac1 expression in both cellular and tissue lysates (including when stimulated with H2O2, interferon, or during viral infection). Similarly, we do not observe reproducible proteins that pulldown with Mac1 when assayed by mass spectrometry. It is quite likely that these issues are a result of tissue/sample preparation that results in a loss of the ADPr modification during preparation (especially for acidic residue modifications). This also explains the reliance on IF assays in the PARP14 literature. A very good discussion of these issues is also contained in this paper: https://doi.org/10.1042/BSR20240986.

Nonetheless we have attempted one final experiment. Here, we have measured ADPr modification of cellular lysates upon uninfected conditions as well as upon infection with either WT or N40D mutant virus. For all conditions, this was done with or without treatment of cells with 100 μM of AVI-4206. Measurement of ADPr modifications by western blot using a pan-ADPr antibody revealed a single prominent band with a molecular weight of ~130kDa, that showed a uniform increase in signal upon treatment of cells with AVI-4206 regardless of infection status. While this general trend was also observed with the mono-ADPr antibody, it was not statistically significant in its regulation upon AVI-4206 treatment. We suspect that the major band observed in these western blots is PARP1, as upon enrichment of ADPr proteins from these lysates by Af1521 immunoprecipitation, we find PARP1 to be among the most abundant proteins detected within this molecular weight range. We note that there is a baseline increase in polyADPr detection upon infection of virus with WT Mac1 (relative to uninfected and virus with N40D) and further increase when treated with AVI-4206. This compound-dependent increase is paralleled in the uninfected and N40D conditions. The counterintuitive increase upon WT Mac1 virus infection, which should erase ADPr marks, and the compound-dependent increase in the uninfected condition suggest that there are many indirect effects on ADPr signalling dynamics in this experiment. These results are difficult to reconcile with the specificity profiling of AVI-4206 (Supplementary Figure5: Thermal proteome profiling in A549 cellular lysates). As mentioned above, the lack of consistent signal across reagents for ADPr detection and the timing of monitoring ADPr levels are additional complicating factors.

We added to the results:

“However, we observed no strong consistent signals of global pan-ADP-ribose (panADPr) or mono-ADP-ribose (monoADPr) accumulation in infected cells treated with AVI-4206 in immunoblot analyses (Supplementary Figure 8).”

Methods for experiment:

Calu3 cells were obtained from ATCC and cultured in Advanced DMEM (Gibco) supplemented with 2.5% FBS, 1x GlutaMax, and 1x Penicillin-Streptomycin at 37°C and 5% CO_2_. 5x10^6^ cells were plated in 15-cm dishes and media was changed every 2-3 days until the cells were 80% confluent. The cells were treated with INFy 50 ng/mL (R&D Systems) w/without AVI-4206 100 μM. After 6 hours, the cells were infected with WA1 or WA1 NSP3 Mac1 N40D at a multiplicity of infection (MOI) of 1 for 36 hours. The cells were washed with PBS x 3 and scraped in Pierce IP Lysis Buffer (ThermoFisher) containing 1x HALT protease and phosphatase inhibitor mix (ThermoFisher) on ice. The lysate was stored at -80C until further processing.

The cell lysate was incubated for 5 minutes at room temperature with recombinant benzonase. Following incubation, the lysate was centrifuged at 13,000 rpm at 4°C for 20 minutes, and the supernatant was collected. The samples were then boiled for 5 minutes at 95°C in 1x NuPAGE LDS sample buffer (Invitrogen) with a final concentration of 1X NuPAGE sample reducing agent (Invitrogen). For the detection of ADPr levels in whole-cell lysates, the samples were subjected to SDS-PAGE and Immunoblotting. All primary and secondary antibodies (pan-ADP-ribose antibody (MABE1016, Millipore)), Mono-ADP-ribose antibody (AbD33204, Bio-Rad), HRP-conjugated (Cell signaling), used at a 1:1000 dilution were diluted in 5% non-fat dry milk in TBST. Signals were detected by chemiluminescence (Thermo) and visualized using the ChemiDoc XRS+ System (Bio-Rad). Densitometric analysis was performed using Image Lab (Bio-Rad). Quantification was normalized to Actin. The data are expressed as mean ± SD. Statistical differences were determined using an unpaired t-test in GraphPad Prism 10.3.1.

(2) SARS-CoV-2 escape mutants for AVI-4206 should be generated, sequenced, and evaluated for both ADP-ribosyl hydrolase activity and their susceptibility to inhibition by AVI-4206.

We thank the reviewer for this suggestion. These are indeed key experiments which are currently hampered by the lack of a cell line that is fully responsive to drug treatment. Although infected organoids and macrophages show an effect in response to AVI-4206, viral levels are ~3 logs lower than in cell lines and difficult to sequence. In the absence of a system that would allow meaningful screening for outgrowth of resistant viruses, we have conducted mass spectrometry studies that showed that Mac1 is the only significant hit for AVI-4206 (SupplementaryFigure 5). The suggested outgrowth experiments will be conducted once a responsive cell line model has been established.

(3) Given that Mac1 is found in several coronaviruses, it would be insightful for the authors to test a selection of Mac1 homologs from divergent coronaviruses to assess whether AVI-4206 can inhibit their activity in vitro.

As mentioned above, inconsistencies in ADPr staining limit our ability to directly measure cellular activity. As an alternative approach to measure AVI-4206 selectivity in cells, we have adapted our CETSA assay for SARS-1 and MERs macrodomain proteins and find evidence that AVI-4206 can shift the melting temperature of both proteins, albeit to a lesser degree than that seen for Mac1. In line with MERS being more structurally divergent than SARS-1 from SARS CoV2, the ΔTagg for SARS-1 and MERS are 4℃ and 1℃, respectively, compared to 9℃ for Mac1. These data have been added as Supplementary Fig S3C. Development of broader spectrum pan-inhibitors is on our radar for future work which will more thoroughly assess homologs from divergent coronaviruses.

We added the following sentence to the main results:

“Encouragingly, we were also able to adapt our CETSA assay for SARS-1 and MERs macrodomain proteins and find that AVI-4206 can shift the melting temperature of both proteins, albeit to a lesser degree than that seen for Mac1 (Supplementary Figure 3C).”

We also added this supplementary figure 3:

Minor(1) Line 88, "respectively.heir potency"

Fixed, thank you!

(2) Line 149 add a period after proteome

Fixed, thank you!

**Reviewer #2 (Recommendations for the authors):**
(a) The authors assess inhibition of MacroD2 and Targ1 as of-targets for AVI-4206. However, Mac1 belongs to the MacroD-type class of macrodomains of which MacroD1, MacroD2 and MOD1s of PARP9 and PARP14 are the human members. In contrast Targ1 belongs to the ALC1-like class, which is only very distantly related to Mac1. Furthermore, recent studies have shown that the first macrodomains of PARP9 and PARP4 (MOD1 of PARP9/14) are much closer related to Mac1 and PARP9/14 were implicated in antiviral immunity. As such the authors should include assays showing the activity of their compounds against MacroD1 and MOD1s of PARP9/14.

We emphasize that we detect no significant shift for any protein other than Mac1 in A549 cells by CETSA-MS (Supplementary Figure 6). For Mac1 CESTA, we see an average of 6 PARP14 spectral counts across conditions and did not detect PARP9. In addition, for separate work in MPro, we ran similar CETSA experiments where we observed an average of 2 PARP9 and 15 PARP14 spectral counts across conditions. Although PARP9 and PARP14 massively increase expression upon IFN treatment in A549 cells, both proteins have been detected by Western Blot in A549 cells previously at baseline.

Nonetheless, we have included modeling of more diverse macrodomains as a supplemental figure and added to the text:

Modeling of other diverse macrodomains, including those within human PARP9 and PARP14 further suggests that AVI-4206 is selective for Mac1 (Supplementary Figure 4).

(b) In the context of SARS-CoV-2 superinfection are a known major complication of infections. These superinfections are associated with lung damage and therefore it would be good if the authors could assess lung damage, e.g. by histology, to see if their treatment has a positive impact on lung damage and thus may help to suppress complications.

We performed histology and the results are inconclusive, but suggest that AVI-4206 treatment could lower apoptosis.There is no difference in pathology between the N40D cohort and vehicle with these markers. This could suggest that AVI-4206 provides an additional mechanism that results in protection. We added to the results:

Caspase 3 staining shows that AVI-4206 treatment reduces apoptosis in the lungs compared to vehicle controls. Additionally, Masson's Trichrome staining reveals a significant reduction in collagen deposition, a surrogate for lung pathology, in the lungs of AVI-4206 treated animals (Supplementary Figure 9).

Histology:

Mouse lung tissues were fixed in 4% PFA (Sigma Aldrich, Cat #47608) for 24 hours, washed three times with PBS and stored in 70% ethanol. All the stainings were performed at Histo-Tec Laboratory (Hayward, CA). Samples were processed, embedded in paraffin, and sectioned at 4μm. The slides were dewaxed using xylene and alcohol-based dewaxing solutions. Epitope retrieval was performed by heat-induced epitope retrieval (HIER) of the formalin-fixed, paraffin-embedded tissue using citrate-based pH 6 solution (Leica Microsystems, AR9961) for 20 mins at 95°C. The tissues were stained for H&E, caspase-3 (Biocare #CP229c 1:100), and trichrome, dried, coverslipped (TissueTek-Prisma Coverslipper), and visualized using Axioscan 7 slide scanner (ZEISS) at 40X. Image quantification was performed with Image J software and GraphPad Prism.

(c) Fig. 1D labelling is wrong

Thank you - fortunately the data were plotted correctly and it was just the inset table of values that was incorrect. This is now fixed!

(d) Line 88: "T" missing at start of sentence

Fixed, thank you!

(e) Line 118: NudT5/AMP-Glo assay was developed in https://doi.org/10.1021/acs.orglett.8b01742

We have added this foundational reference, thank you!

(f) Line 147ff: It would be good if the authors could highlight that the TPP methodology has known limitations (e.g. detection of low abundance proteins and low thermal shift of some binders) and thus is not an absolute proof that AVI-4206 "engage with high specificity for Mac1"

We added this important context to the concluding sentence of this paragraph:

“While this assay may not be sensitive to detection of proteins with low abundance proteins or low thermal shift upon ligand binding, collectively, these results indicate that AVI-4206 can cross cellular membranes and engage with high specificity for Mac1.”

(g) The authors use their well established in vitro Mac1 model as well as the SARS-CoV-2 WA strain. Given the ongoing diversification of SARS-CoV-2 and the current prevalence of the Omicron VOC it would be good if the authors could investigate whether alteration in Mac1 occurred or are detected which could influence the efficacy of their inhibitor. Similarly, it would be interesting to know how effective their drug is on other clinically relevant beta-CoV Mac1, e.g. from MERS or SARS1.

We thank the reviewer for the suggestion. Mac1 is one of the more conserved areas of the SARS-CoV-2 genome as there has only been one nonsynonymous mutation V34L (Orf1a:V1056L) that recently emerged in the BA.2.86 lineage and is now in all of the JN.1 derivatives. Currently, the mutation is only ~80% penetrant in circulating SARS-CoV-2 sequences suggesting that it might revert to wild-type and is not associated with a fitness benefit. Based on our structural analysis (shown in Supplementary Figure4D above), we do not believe this mutation affects AVI-4206 binding, but we are including this variant in our future in vitro and in vivo studies as well as other beta-CoV. For SARS and MERS, see response to Reviewer 1 using CETSA to show that these targets are engaged by AVI-4206.

(h) As methods to detect PARP14-derived ADP-ribosylation are available and it was shown that Mac1 can reverse this modification in cells. It would be good if the authors could investigate the impact of AVI-4206 on ADP-ribosylation in vivo.

To test this idea we adapted the IF assay used by others in the field and show an effect of AVI-4206. We have added to the text:

Although the IFN response was not sufficient to control viral replication, it is possible that the changes in ADP-ribosylation, in particular marks catalyzed by PARP14, downstream of IFN treatment could serve as a marker for Mac1 efficacy (Ribeiro et al. 2025). To investigate whether downstream signals from PARP14 were specifically erased by Mac1, we used an immunofluorescence assay that showed that Mac1 could remove IFN-γ-induced ADP-ribosylation that is mediated by PARP14 (Kar et al. 2024). We stably expressed wild-type Mac1 and the N40D mutant Mac1 in A549 cells. The data showed that Mac1 expression decreased IFN-γ-induced ADP-ribosylation, whereas the Mac1-N40D mutant did not (Figure 3E, F), indicating that Mac1 mediates the hydrolysis of IFN-γ-induced ADP-ribosylation. The PARP14 inhibitor RBN012759 completely blocked IFN-γ-induced ADP-ribosylation (Figure 3E, F), further confirming that IFN-γ-induced ADP-ribosylation is mediated by PARP14. AVI-4206 reversed the Mac1-induced hydrolysis of ADP-ribosylation and enhanced the ADP-ribosylation signal in Mac1-overexpressing cells (Figure 3E, F), further demonstrating its ability to inhibit the hydrolase activity of Mac1. We further validated this result using different ADP-ribosylation antibodies for immunofluorescence (Supplementary Figure 7). However, we observed no strong consistent signals of global pan-ADP-ribose (panADPr) or mono-ADP-ribose (monoADPr) accumulation in infected cells treated with AVI-4206 in immunoblot analyses (Supplementary Figure 8). Collectively, these results provide further evidence that simple cellular models are insufficient to explore the effects of Mac1 inhibition and that monitoring specific PARP14-mediated ADP-ribosylation patterns can provide an accessible biomarker for the efficacy of Mac1 inhibition.

A549 Mac1 expression cell construction

Mac1 wild-type (Mac1) and N1062D mutant (Mac1 N1062D) gene fragments were loaded into pLVX-EF1α-IRES-Puro (empty vector, EV) using Gibson cloning kit (NEB E5510). Lentivirus was prepared as previously described (PMID: 30449619; DOI: 10.1016/j.cell.2018.10.024). Briefly, 15 million HEK293T cells were grown overnight on 15 cm poly-L-Lysine coated dishes and then transfected with 6 ug pMD2.G (Addgene plasmid # 12259 ; http://n2t.net/addgene:12259 ; RRID:Addgene_12259), 18 ug dR8.91 (since replaced by second generation compatible pCMV-dR8.2, Addgene plasmid #8455) and 24 ug pLVX-EF1α-IRES-Puro (EV, Mac1, Mac1-N1062D) plasmids using the lipofectamine 3000 transfection reagent per the manufacturer’s protocol (Thermo Fisher Scientific, Cat #L3000001). pMD2.G and dR8.91 were a gift from Didier Trono. The following day, media was refreshed with the addition of viral boost reagent at 500x as per the manufacturer’s protocol (Alstem, Cat #VB100). Viral supernatant was collected 48 hours post transfection and spun down at 300 g for 10 minutes, to remove cell debris. To concentrate the lentiviral particles, Alstem precipitation solution (Alstem, Cat #VC100) was added, mixed, and refrigerated at 4°C overnight. The virus was then concentrated by centrifugation at 1500 g for 30 minutes, at 4°C. Finally, each lentiviral pellet was resuspended at 100x of original volume in cold DMEM+10%FBS+1% penicillin-streptomycin and stored until use at -80°C. To generate Mac1 overexpressing cells, 2 million A549 cells were seeded in 10 cm dishes and transduced with lentivirus in the presence of 8 μg/mL polybrene (Sigma, TR-1003-G). The media was changed after 24h and, after 48 hours, media containing 2μg/ml puromycin was added. Cells were selected for 72 hours and then expanded without selection. The expression of Mac1 was confirmed by Western Blot.

Immunofluorescence assay:

To assess the effect of Mac1 on IFN-induced ADP-ribosylation. A549-pLVX-EV, A549-pLVX-Mac1 and A549-pLVX-Mac1-N1062D cells were seeded in 96-well plate (10,000 cells/well). Cells were pre-treated with medium or 100 unit/mL IFN-γ (Sigma, SRP3058) for 24 hours to induce the expression of ADP-ribosylation. These 3 cell lines were then treated the next day with the indicated concentrations of AVI-4206 or RBN012759 (Medchemexpress, HY-136979). After 24 hours of exposure to drugs, treated cells were fixed in pre-cooled methanol at -20°C for 20 min, blocked in 3% bovine serum albumin for 15 min, incubated with Poly/Mono-ADP Ribose (E6F6A) Rabbit mAb (CST, 83732S) or Poly/Mono-ADP Ribose (D9P7Z) Rabbit mAb (CST, 89190S) antibodies for 1 h, and then incubated with Goat anti-Rabbit IgG Secondary Antibody, Alexa Fluor 488 (ThermoFisher, A-11008) secondary antibodies for 30 min and stained with DAPI for 10 minutes. Fluorescent cells were imaged with an IN Cell Analyzer 6500 System (Cytiva) and analyzed using IN Carta software (Cytiva).

**Reviewer #3 (Recommendations for the authors):**
Just a couple of observations/details that might help strengthen the article:(1) The caco-1 data for AVI4206 would suggest that there is some sort of efflux going on, yet there is no mention of it in the paper. This might be useful in the optimization paradigm moving forward.

We thank the reviewer for this observation and suggestion. Indeed, we believe that efflux is behind the low oral bioavailability of AVI-4206. We are working specifically to remove this liability in next-generation analogs, using the caco2 assay to guide this ongoing effort. Keep an eye out for a preprint on this soon! We have added to the discussion:

“In addition to dissecting such molecular mechanisms of macrodomain function and inhibition, future efforts will focus on improving pharmacokinetic properties, including a cellular efflux liability that results in low oral bioavailability of AVI-4206. ”

(2) There are some spectroscopic anomalies/mistakes in the NMR data. The carbon NMR for 1-((8-amino-9H-pyrimido[4,5-b]indol-4-yl)amino)pyrrolidin-2-one should only have 14 unique carbons, but the authors report 15. The HNMR for AVI1500 should only have 19 H's, but the authors list 20. The HNMR data for AVI3762/3763 should have 16 H's, but the authors only report 13. The CNMR for AVI4206 should only have 19 unique carbons, but the authors report 20.

Thank you for noting these inconsistencies regarding the reported NMR spectra. We have rectified them by more closely examining the spectra and in some cases acquiring new data. We identified one peak (47.9) in the 13C NMR of 1-((8-amino-9H-pyrimido[4,5-b]indol-4-yl)amino)pyrrolidin-2-one that is apparently an artifact of the automated peak picking in the data analysis software. In the 1H NMR of AVI-1500, the triplet peak at 7.20 integrates to 1H, but was erroneously reported as 2H in the original manuscript. This error has been corrected. Spectra were re-acquired for AVI-3762, AVI-3763, and AVI-4206 with longer acquisition times, and/or on a 600 MHz spectrometer to afford the complete line lists now reported in the revised manuscript. Please note AVI-4206 has 18 distinct 13C resonances due to the equivalence of the gem-dimethyl methyl groups.